

# Electronic structure of liquid xenon in the context of light dark matter direct detection

Riccardo Catena[1*], Luca Marin[2†], Marek Matas[2,3‡],
Nicola A. Spaldin[2∘] and Einar Urdshals[1§]

**1** Chalmers University of Technology, Department of Physics, SE-412 96 Göteborg, Sweden
**2** Materials Theory, ETH Zürich, Wolfgang-Pauli-Strasse 27, 8093 Zürich, Switzerland
**3** Faculty of Nuclear Sciences and Physical Engineering,
Czech Technical University in Prague, Czech Republic

* riccardo.catena@chalmers.se , † luca.marin@mat.ethz.ch , ‡ marek.matas@fjfi.cvut.cz ,
∘ nspaldin@ethz.ch , § einarurdshals@gmail.com

## Abstract

We present a description of the electronic structure of xenon within the density-functional theory formalism with the goal of accurately modeling dark-matter-induced ionisation in liquid xenon detectors. We compare the calculated electronic structures of the atomic, liquid and crystalline solid phases, and find that the electronic charge density and its derivatives in momentum space are similar in the atom and the liquid, consistent with the weak interatomic van der Waals bonding. The only notable difference is a band broadening of the highest occupied $5p$ levels, reflected in the densities of states of the condensed phases, as a result of the inter-atomic interactions. We therefore use the calculated density of states of the liquid phase, combined with the standard literature approach for the isolated atom, to recompute ionisation rates and exclusion limit curves for the XENON10 and XENON1T experiments. We find that the broadening of the $5p$ levels induced by the liquid phase is relevant only for dark matter masses below 6 MeV, where it increases the ionisation rate relative to that of the isolated atom. For most of the explored mass range, the energies of the discrete $4d$ and $5s$ levels have the strongest effect on the rate. Our findings suggest a simple scheme for calculating dark matter-electron scattering rates in liquid noble gas detectors, using the calculated values for the atom weighted by the density of states of the condensed phase.



# 1  Introduction

According to the current cosmological standard model, more than 80% of matter in the universe is so-called dark matter (DM) [1]. DM exerts a gravitational pull on galaxies and stars and plays a crucial role in the large-scale structure formation of the universe. Although its indirect effects have been demonstrated across a wide range of measurements, non-gravitational detection of dark-matter particles [2] has not yet been achieved.

Until recently, one of the most extensively investigated candidates for DM has been the weakly interacting massive particle (WIMP) [3]. WIMPs are proposed particles with interactions at the weak scale that originally arose from theories outside of the area of DM, such as supersymmetry [4, 5]. The expected mass range of WIMPs is between the GeV and few hundreds of TeV energy scale, which is comparable to the mass of atomic nuclei [4]. Elastic scattering of WIMPs with atomic nuclei should therefore be kinematically favourable, motivating large-scale experiments to detect the possible signal coming from individual nuclear recoils. In particular, earth-based detectors containing liquid xenon as the target material have achieved the highest sensitivity to spin-independent DM interactions in the WIMP mass range [6,7]. The lack of an unambiguous detection, however, is now motivating a search outside of the WIMP paradigm, particularly in regions of smaller DM particle masses in the range between the MeV and the GeV [8]. This is the mass window where known standard model particles, such as protons, neutrons and electrons, lie, and where the present DM cosmological density could be explained by the chemical decoupling from standard model matter of a DM species that couples to visible matter through beyond-the-standard-model interactions [8].

While nuclear recoil experiments are used to search for WIMPs, DM particles lighter than WIMPs must be sought in the recoils of lighter targets such as electrons [9]. Since large volumes of high-purity liquid xenon are already available in existing detectors, they have been progressively repurposed to detect DM-electron interactions together with nuclear recoils [10–13]. The possibility of detecting sub-GeV DM with xenon detectors is based on the fact that a non-relativistic particle heavier than a few MeV carries enough kinetic energy to ionise xenon atoms in a medium [9]. To guide such an effort, an accurate calculation of the sensitivity of these detectors is crucial. In particular, reliable estimates of the expected event rates, DM-mass reach and exclusion limits for DM-electron scattering are needed.

To calculate these properties, precise knowledge of the electronic wave functions and energies in liquid xenon is essential, as these are key ingredients in the quantum mechanical modeling of the DM-electron interaction [14, 15]. So far, the modeling of DM-electron interactions in liquid xenon detectors has assumed that atoms can be treated as being isolated [10, 16], neglecting the impact that the liquid phase has on the electronic wave functions and binding energies.

Here, we develop a framework that describes the electronic structure of xenon in its condensed phases (solid and liquid), with a particular focus on the liquid due to its use in current detectors. Our method of choice is density functional theory (DFT), whose combination of chemical accuracy and computational affordability makes it the leading choice for the calculation of DM-electron scattering processes in materials [9, 15, 17–22]. Based on the theorems of Hohenberg and Kohn [23], DFT minimizes the electronic energy, which is a functional of the charge density, in a variational procedure to obtain the energy and charge density of the ground state. Using an approximation proposed by Kohn and Sham [24], a set of simultaneous single-particle eigenvalue equations, with the many-body exchange and correlation interactions mapped into an additional effective potential, is solved self-consistently. This procedure is conveniently implemented in computer codes, and the resulting electron density completely describes the ground-state properties of the system, within the approximation chosen for the exchange-correlation term.

We proceed by first observing, in Sec. 2, that the DM-electron scattering rate can be evaluated from two main physical observables: i) the electron density in momentum space and its gradients and ii) the binding energies of the electrons, which are described by the density of states in the condensed phases. Since DFT is a convenient tool for computing these quantities for all the phases, including the liquid, we next determine the most appropriate flavour of DFT, both in terms of fundamental physical approximations - in particular, the choice of exchange-correlation functional - and in terms of computational parameters for modeling the various phases of xenon (Sec. 3). Next, in Secs. 4.1 and 4.2, we compare the calculated electron densities and density gradients in momentum space for the atom and the liquid, to visualize the impact of the condensation on the DM-electron scattering process. We also compare the DFT atomic charge densities with those obtained from the standard Roothan-Hartree-Fock (RHF) choice for modeling initial electronic states in noble gases [10, 16]. Based on these comparisons, we propose an efficient scheme for calculating DM-electron scattering rates in liquid xenon detectors in which all the information about the liquid phase is carried by the density of states. Finally, we use our scheme to quantify the effects of the liquid phase on exclusion limits from XENON10 and XENON1T experimental data (Sec. 4.3).

## 2 Xenon and dark matter direct detection

In this section, we introduce a general notation that enables us to describe the rate of DM-induced ionisation in atomic or liquid xenon in a unified manner, and is useful in highlighting

the main differences and similarities between the two xenon phases. In Part A, we write the ionisation rate in a form that is independent of the choice of initial and final states (labelled {1} and {2}), while in Part B, we show that this rate can be written in terms of the electron density and its gradients in momentum space.

## 2.1 Dark matter-induced ionisation in xenon detectors

The rate at which an isolated xenon atom or a liquid xenon sample can be ionised by DM-electron interactions is [16],

$$\mathscr{R}_{1\to2} = \int \frac{d^3\mathbf{q}}{(2\pi)^3} \int d^3\mathbf{v} f_\chi(\mathbf{v}) (2\pi)\delta(E_2 - E_1 + \Delta E_\chi) \frac{n_\chi}{16m_\chi^2 m_e^2} \overline{|\mathcal{M}_{1\to2}|^2}. \tag{1}$$

Here, we adopt a general notation where {1} collectively denotes the quantum numbers of the initial state electron bound to the isolated xenon atom or liquid xenon sample, while {2} describes the quantum numbers of the electron ejected in the ionisation process $1 \to 2$. In Eq. (1), $\mathbf{v}$ is the incoming DM particle velocity, $f_\chi(\mathbf{v})$ is the local DM velocity distribution in the detector rest frame, $\Delta E_\chi \equiv q^2/(2m_\chi) - \mathbf{v} \cdot \mathbf{q}$ is the difference between the outgoing and incoming DM particle energy, $\mathbf{q}$ is the momentum transfer in the ionising DM-electron interaction, $n_\chi$ is the local DM number density, and $m_\chi$ and $m_e$ are the DM and electron mass, respectively. For $f_\chi(\mathbf{v})$, we assume a truncated Maxwell-Boltzmann distribution with most probable speed $v_0 = 238$ km s$^{-1}$ [25], galactic escape speed $v_{\text{esc}} = 544$ km s$^{-1}$ [25] and Earth's speed in a reference frame with zero mean DM particle velocity given by $v_e = 250.5$ km s$^{-1}$ [25]. We calculate $n_\chi \equiv \rho_\chi/m_\chi$ assuming $\rho_\chi = 0.4$ GeV cm$^{-3}$ [26]. Finally, we denote by $E_1$ ($E_2$) the initial (final) state electron energy, and by $\overline{|\mathcal{M}_{1\to2}|^2}$ the squared modulus of the DM-electron scattering amplitude summed (averaged) over final (initial) DM and electron spin states. We assume that it is given by [19]

$$\overline{|\mathcal{M}_{1\to2}|^2} = c_1^2 |F_{\text{DM}}(q)|^2 |f_{1\to2}(\mathbf{q})|^2, \tag{2}$$

where $c_1$ is a coupling constant,[1] and $F_{\text{DM}}(q)$ is the so-called "DM form factor", associated with the range of the underlying DM-electron interaction. For example, for interactions mediated by a heavy particle, $F_{\text{DM}}(q) = 1$, whereas for DM-electron interactions associated with a light mediator, $F_{\text{DM}}(q) = q_{\text{ref}}^2/q^2$, where $q_{\text{ref}} = \alpha m_e$ is a reference momentum, and $\alpha$ is the fine structure constant. Within the non-relativistic effective theory of DM-electron interaction [16], Eq. (2) arises when the underlying DM-electron interaction is described by the quantum mechanical operator $\mathcal{O}_1 = \mathbb{1}_e \mathbb{1}_\chi$, where $\mathbb{1}_e$ and $\mathbb{1}_\chi$ are the identities in the electron and DM spin spaces, respectively. Eq. (2) also arises from the non-relativistic reduction of the so-called dark photon model [27].

In Eq. (2), $f_{1\to2}(\mathbf{q})$ is the overlap integral between the initial and final state electron wave functions, and is given by

$$f_{1\to2}(\mathbf{q}) = \int \frac{d^3\mathbf{k}}{(2\pi)^3} \widetilde{\psi}_2^*(\mathbf{k}) \widetilde{\psi}_1(\mathbf{k}-\mathbf{q}), \tag{3}$$

where $\widetilde{\psi}_1$ ($\widetilde{\psi}_2$) is the initial (final) state electron wave function in $\mathbf{k}$ space. Notice that $\mathscr{R}_{1\to2}$ in Eq. (1) gives the rate for the specific ionisation process corresponding to initial and final states labelled by the quantum numbers generically denoted by {1} and {2}. The total ionisation rate is therefore

$$\mathscr{R} = 2 \sum_{\{1\}\{2\}} \mathscr{R}_{1\to2}, \tag{4}$$

---

[1]Note that here the label 1 refers to a specific DM-electron coupling constant and not to the initial state quantum numbers.

where the factor of 2 accounts for the electron spin degeneracy.

Inspection of Eq. (1) shows that the double sum in Eq. (4) acts only on the product between the energy conservation Dirac delta and the modulus squared of the electron wave function overlap integral $f_{1\rightarrow2}(\mathbf{q})$. Consequently, it is convenient to introduce the following function

$$\Delta(\mathbf{q},\mathbf{v}) = \sum_{\{1\}\{2\}} (2\pi)\delta(E_2 - E_1 + \Delta E_\chi(\mathbf{q},\mathbf{v}))|f_{1\rightarrow2}(\mathbf{q})|^2 \,, \tag{5}$$

and rewrite the rate $\mathscr{R}$ in the following compact form,

$$\mathscr{R} = \frac{c_1^2 n_\chi}{8m_\chi^2 m_e^2} \int \frac{d^3\mathbf{q}}{(2\pi)^3} \int d^3\mathbf{v} f_\chi(\mathbf{v})|F_{\text{DM}}(q)|^2 \Delta(\mathbf{q},\mathbf{v}) \,. \tag{6}$$

*Ionisation in the atomic limit.* When the initial state electron wave function describes an atomic orbital with principal, orbital and magnetic quantum numbers denoted by $n$, $\ell$ and $m$, respectively, and the final state electron wave function is a positive energy solution of the atomic Schrödinger equation, $\psi_{k'\ell'm'}$ (defined below in Eq. (26)) the sum in Eq. (4) reads as follows,

$$\sum_{\{1\}\{2\}} = \sum_{n\ell m}\sum_{k'\ell'm'} = \frac{1}{4\pi}\sum_{n\ell m}\sum_{\mathbf{k}'\ell'm'} \quad\rightarrow\quad \frac{1}{4\pi}\sum_{n\ell m}\sum_{\ell'm'}\int\frac{V d^3\mathbf{k}'}{(2\pi)^3} \,, \tag{7}$$

where $k' = |\mathbf{k}'|$ and $k'^2/(2m_e)$ is the outgoing electron kinetic energy.

By writing the initial- and final-state electron energies as $E_1 = E_{n\ell} - \Phi$ and $E_2 = k'^2/(2m_e)$, for $\Delta(\mathbf{q},\mathbf{v})$ we find

$$\Delta(\mathbf{q},\mathbf{v}) = \frac{\pi}{2}\sum_{n\ell}\int\frac{dk'}{k'}\delta\left(\frac{k'^2}{2m_e} + \Phi - E_{n\ell} + \Delta E_\chi(\mathbf{q},\mathbf{v})\right)W^{n\ell}(k',q) \,. \tag{8}$$

Here $E_{n\ell} \leq 0$ is the energy of the $(n,\ell)$ orbital relative to the highest occupied energy level of xenon, which lies at energy $-\Phi$, where $\Phi$ is the ionisation potential, that is, the energy difference between the highest occupied energy level and the vacuum level at 0 eV. For the case of the isolated Xe atom, $\Phi$ is known experimentally to be 12.1 eV [28]. Furthermore [16], the material response function can be defined as

$$W^{n\ell}(k',q) = V\frac{4k'^3}{(2\pi)^3}\sum_m\sum_{\ell'm'}|f_{n\ell m\rightarrow k'\ell'm'}(\mathbf{q})|^2 \,. \tag{9}$$

*Ionisation in the condensed phase with periodic boundary conditions.* When periodic boundary conditions (PBC) apply, the initial state electron can be described by a Bloch state. In this case, and assuming a plane wave as the final state, we can rewrite the double sums in Eq. (4) as integrals by taking the continuous limit:

$$\sum_{\{1\}\{2\}} = \sum_{i\mathbf{K}}\sum_{\mathbf{k}'} \quad\longrightarrow\quad \sum_i\int_{\text{BZ}}\frac{V d^3\mathbf{K}}{(2\pi)^3}\int\frac{V d^3\mathbf{k}'}{(2\pi)^3} \,. \tag{10}$$

Here $V$ is a finite normalisation volume, $\mathbf{k}'$ is the final-state plane-wave momentum, $i$ is the initial-state band index and $\mathbf{K}$ is the wave vector in the first Brillouin Zone (BZ).

In this situation, with $\psi_1 = \psi_{i\mathbf{K}}$ (Bloch state), $\psi_2 = \psi_{\mathbf{k}'}$ (plane wave) and with initial and final-state electron energies given by $E_1 = E_i(\mathbf{K}) - \Phi$ and $E_2 = k'^2/(2m_e)$, we find

$$\Delta(\mathbf{q},\mathbf{v}) = N_{\text{cell}}\int d^3\mathbf{k}'\int dE_e\, W(\mathbf{k}'-\mathbf{q},E_e)(2\pi)\delta\left(\frac{k'^2}{2m_e} + \Phi - E_e + \Delta E_\chi\right), \tag{11}$$

where [19],

$$W(\mathbf{k}' - \mathbf{q}, E_e) = \frac{V_{\text{cell}}}{(2\pi)^3} \sum_i \int_{\text{BZ}} \frac{d^3\mathbf{K}}{(2\pi)^3} \, \delta\left(E_e - E_i(\mathbf{K})\right) \left|\widetilde{\psi}_{i\mathbf{K}}(\mathbf{k}' - \mathbf{q})\right|^2. \tag{12}$$

Again, $\Phi$ is the energy difference between the highest occupied energy level, which in this case is the top of the valence band density of states, and the vacuum level at 0 eV. We note that $\Phi$ is distinct from the work function, which is the energy needed to remove an electron from the condensed phase, and also contains the energy cost needed to overcome any barrier at the surface. Here, we have rewritten the normalisation volume $V$ as the product between the unit cell volume, $V_{\text{cell}}$, and the number of unit cells in the xenon target, $N_{\text{cell}}$. This factorization is relevant in the DFT calculations presented below.

The function $W(\mathbf{k}' - \mathbf{q}, E_e)$ is normalised as follows

$$\int dE_e \int d^3(\mathbf{k}' - \mathbf{q}) \, W(\mathbf{k}' - \mathbf{q}, E_e) = N_{\text{bands}}, \tag{13}$$

where $N_{\text{bands}}$ is the number of occupied bands included in the calculation.

## 2.2 Ionisation rate and electron density in xenon detectors

We now use the formalism introduced in Part A to relate the rate of DM-induced ionisations to the *electron density* in a xenon target, assuming that the ejected electron is described by a wave function that has a narrow distribution around a specific momentum. The description in terms of the electron density will be important when comparing predictions obtained for different xenon phases.

When the final state wave function $\widetilde{\psi}_2$ peaks at a definite momentum $\mathbf{k}'$, and the initial state wave function $\widetilde{\psi}_1$ varies slowly around $\mathbf{k}'$, we can expand $f_{1\to 2}$ in Eq. (3) as

$$f_{1\to 2}(\mathbf{q}) = \int \frac{d^3\mathbf{k}}{(2\pi)^3} \, \widetilde{\psi}_2^*(\mathbf{k}) \left[ \widetilde{\psi}_1(\mathbf{k}' - \mathbf{q}) + \nabla_{\mathbf{k}} \widetilde{\psi}_1(\mathbf{k} - \mathbf{q})|_{\mathbf{k}=\mathbf{k}'} \cdot (\mathbf{k} - \mathbf{k}') + \dots \right]. \tag{14}$$

When the final state electron is described by a plane wave and thus $\widetilde{\psi}_2(\mathbf{k}) = \delta(\mathbf{k} - \mathbf{k}')/\sqrt{V}$, only the first term contributes to the expansion in Eq. (14). Consequently, the gradient terms in Eq. (14) are associated with deviations from the plane-wave approximation for the final-state electron wave function $\widetilde{\psi}_2(\mathbf{k})$.

As long as the expansion in Eq. (14) holds, the squared modulus of $f_{1\to 2}$, and thus the total ionisation rate, can be related to the initial density of electrons in the target and its gradients. To show this, let us decompose $\Delta(\mathbf{q}, \mathbf{v})$ as

$$\Delta(\mathbf{q}, \mathbf{v}) = \sum_{n\ell} \Delta^{n\ell}(\mathbf{q}, \mathbf{v}), \tag{15}$$

where

$$\Delta^{n\ell}(\mathbf{q}, \mathbf{v}) \equiv \int \frac{dE_e}{2\pi} \sum_{\{1\to n\ell\}} \sum_{\{2\to n\ell\}} |f_{1\to 2}(\mathbf{q})|^2, \tag{16}$$

with

$$\sum_{\{1\to n\ell\}} = \sum_{\{1\}} (2\pi)\delta(E_e - E_1 - \Phi)\,\theta(E_e - E_{\min}^{n\ell})\theta(E_{\max}^{n\ell} - E_e), \tag{17}$$

and

$$\sum_{\{2\to n\ell\}} = \sum_{\{2\}} (2\pi)\delta(E_2 - E_e + \Phi + \Delta E_\chi). \tag{18}$$

These equations apply both to the case of isolated atoms, where $\psi_1 = \psi_{n\ell m}$ is an atomic orbital, and to the cases of solid or liquid xenon, where $\psi_1 = \psi_{i\mathbf{K}}$ is a Bloch state. In the case of isolated atoms, the energy spectrum is discrete, and $E_{\min}^{n\ell} = E_{\max}^{n\ell} = E_{n\ell}$, where $E_{n\ell}$ is the energy eigenvalue of the atomic orbital $\psi_{n\ell m}$. In the cases of solid or liquid xenon, the energy levels broaden into continuous bands, with lower and upper bounds $E_{\min}^{n\ell}$ and $E_{\max}^{n\ell}$. In the above equations, the integration variable $E_e$ is the frequency conjugate to the time coordinate (see footnote 4). In practice, it is an auxiliary variable that we integrate over to guarantee that only energy levels within $\Delta E_e^{n\ell} \equiv E_{\max}^{n\ell} - E_{\min}^{n\ell}$ contribute to $\Delta^{n\ell}(\mathbf{q}, \mathbf{v})$.

By setting $E_e = E_{n\ell}$ in Eq. (18),[2] and inserting Eq. (14) into Eq. (16), we obtain[3]

$$
\begin{aligned}
\Delta^{n\ell}(\mathbf{q}, \mathbf{v}) = {}& \alpha^{n\ell} \rho_e^{n\ell}(\mathbf{k}' - \mathbf{q}) \\
& + \boldsymbol{\beta}^{n\ell} \cdot \nabla_{\mathbf{k}'} \rho_e^{n\ell}(\mathbf{k}' - \mathbf{q}) \\
& + \frac{1}{2} \gamma_{ij}^{n\ell} \nabla_{k_i'} \nabla_{k_j'} \rho_e^{n\ell}(\mathbf{k}' - \mathbf{q}) \\
& + \dots \text{ higher order gradient terms,}
\end{aligned}
\tag{19}
$$

where[4]

$$
\rho_e^{n\ell}(\mathbf{k}' - \mathbf{q}) = 2 \int \frac{\mathrm{d}E_e}{2\pi} \sum_{\{1 \to n\ell\}} |\widetilde{\psi}_1(\mathbf{k}' - \mathbf{q})|^2 \,,
\tag{23}
$$

is the Fourier transform of the density of electrons with quantum numbers $n\ell$, in the atomic case, and with energies $E_i(\mathbf{K})$ in $\Delta E_e^{n\ell}$, in the case of solid or liquid xenon. Finally,

$$
\begin{aligned}
\alpha^{n\ell} &= \frac{1}{2} \sum_{\{2 \to n\ell\}} |\psi_2(0)|^2 \,, \\
\boldsymbol{\beta}^{n\ell} &= \frac{1}{2} \sum_{\{2 \to n\ell\}} \left[ \psi_2^*(0) \hat{\boldsymbol{P}} \psi_2(0) - \mathbf{k}' |\psi_2(0)|^2 \right] \,, \\
\gamma_{ij}^{n\ell} &= \frac{1}{2} \sum_{\{2 \to n\ell\}} \Big[ (\hat{P}_i \psi_2(0))^* \hat{P}_j \psi_2(0) + k_i' k_j' |\psi_2(0)|^2 \\
&\qquad\qquad - k_i' \psi_2^*(0) \hat{P}_j \psi_2(0) + k_j' \psi_2(0) (\hat{P}_i \psi_2(0))^* \Big] \,,
\end{aligned}
\tag{24}
$$

with

$$
\hat{\boldsymbol{P}} = -i \boldsymbol{\nabla} \,.
\tag{25}
$$

---

[2]This is an exact replacement in the case of isolated atoms, and a good approximation when the broadening of atomic energy levels in the solid/liquid phase is small. While we will not use this approximation in our numerical implementations (for these, we will use another approximation explained in Sec. 3.6), here it allows us to disentangle the two sums in the definition of $\Delta^{n\ell}(\mathbf{q}, \mathbf{v})$ for all values of $n$ and $\ell$.

[3]Consistent with Eq. (14), in the derivation of Eq. (19) we neglect second derivatives of the initial state electron wave function.

[4] Notice that

$$
\begin{aligned}
\rho_e^{n\ell}(\mathbf{k}' - \mathbf{q}) &\equiv \rho_e^{n\ell}(\mathbf{k}' - \mathbf{q}, t = 0) \\
&= 2 \sum_{\{1\}} \int_{\Delta E_e^{n\ell}} \frac{\mathrm{d}E_e}{2\pi} \int_{\Delta E_e^{n\ell}} \frac{\mathrm{d}E_e'}{2\pi} \widetilde{\psi}_1(\mathbf{k}' - \mathbf{q}, E_e) \widetilde{\psi}_1^*(\mathbf{k}' - \mathbf{q}, E_e') \,.
\end{aligned}
\tag{20}
$$

By using for the initial state electron wave function at time $t$,

$$
\widetilde{\psi}_1(\mathbf{k}' - \mathbf{q}, t) = e^{-i(E_1 + \Phi)t} \widetilde{\psi}_1(\mathbf{k}' - \mathbf{q}) \,,
\tag{21}
$$

and for its time Fourier transform at $E_e$,

$$
\widetilde{\psi}_1(\mathbf{k}' - \mathbf{q}, E_e) = (2\pi)\delta(E_1 - E_e + \Phi) \widetilde{\psi}_1(\mathbf{k}' - \mathbf{q}) \,,
\tag{22}
$$

we can rewrite the electron density in a xenon detector as in Eq. (23).

Here, $\psi_2$ (without a tilde) is the final-state electron wave function in real space. For a given $\psi_2$, the factors $\alpha^{n\ell}$, $\boldsymbol{\beta}^{n\ell}$ and $\gamma_{ij}^{n\ell}$ can be evaluated explicitly. For example, in the case of a plane wave, one has $|\psi_2(0)|^2 = 1/V$, where $V$ is the normalisation volume. In the case of a positive energy solution of the hydrogen atom Schrödinger equation with effective nuclear charge $Z_{\text{eff}}$, $\psi_2(\mathbf{x})$ is given by

$$\psi_2(\mathbf{x}) = \psi_{k'\ell'm'}(\mathbf{x}) = R_{k'\ell'}(r)\, Y_{\ell'm'}(\theta, \phi)\,, \tag{26}$$

with

$$R_{k'\ell'}(r) = \frac{(2\pi)^{3/2}}{\sqrt{V}}(2k'r)^{\ell'} \frac{\sqrt{\frac{2}{\pi}}\left|\Gamma\left(\ell'+1-i\eta\right)\right| e^{\frac{\pi\eta}{2}}}{(2\ell'+1)!} e^{-ik'r}\,_1F_1\left(\ell'+1+i\eta, 2\ell'+2, 2ik'r\right). \tag{27}$$

Here $r = |\mathbf{x}|$, the angles $\theta$ and $\phi$ identify the direction of $\mathbf{x}$, $\eta = Z_{\text{eff}}/(k'a_0)$, and $a_0 = 1/(\alpha m_e)$ is the Bohr radius. In this case, we find

$$|\psi_2(0)|^2 = \frac{4\pi}{V} F_c(\eta)\, \delta_{0\ell'} \delta_{0m'}\,, \tag{28}$$

where we used $_1F_1(a, b, 0) = 1$ for arbitrary $a$ and $b$, as well as,

$$|\Gamma(1-i\eta)|^2 = \pi\eta / \sinh(\pi\eta)\,. \tag{29}$$

The factor

$$F_c(\eta) = \frac{2\pi\eta}{1 - e^{-2\pi\eta}}\,, \tag{30}$$

in Eq. (28) is the so-called Fermi correction function. By inserting Eq. (19) into Eq. (15), and then the latter into Eq. (6), we obtain an expression for the total ionisation rate as a function of the density $\rho_e^{n\ell}$ and its gradients.

We find this formalism to be convenient in two aspects. In Part A, we wrote a general expression for the rate (Eq. (6)) that is valid independently of the specific form of initial and final states. In Part B, we outlined how to relate the ionisation rate to the initial-state electron density and its gradients in momentum space, which can be used to assess the influence of the different target phases (atomic or liquid) on the ionisation rate.

# 3 Density functional theory description of xenon

For our study, we choose the DFT implementation provided by the plane-wave pseudopotential open-source code `Quantum Espresso v.6.4` [29–31]. This software package and methodology offer a number of practical advantages: Well-tested pseudopotentials, several choices for the exchange-correlation (XC) functional, a number of flavors of van der Waals corrections, and the spin-orbit coupling interaction as well as Hubbard-$U$ corrections for localized states are all implemented. In addition, the use of a plane-wave basis set for the expansion of the Kohn-Sham states is particularly convenient, since convergence of the total energy can be straightforwardly assessed by increasing the number of plane waves. Importantly, `Quantum Espresso` interfaces directly with two leading packages for calculating DM-electron scattering, `QEdark` [15] and its extension `QEdark-EFT` [18, 32]. The specific choice of the pseudopotentials and further computational details are described in Appendix A.

## 3.1 Measured properties of xenon

Xenon is a noble gas with atomic number $Z = 54$, mass number $A \sim 131$ and electronic configuration $[\text{Kr}]\, 4d^{10}\, 5s^2\, 5p^6$. The filled electron shells allow only weak van der Waals interatomic interactions between atoms, so that xenon is gaseous at room temperature and atmospheric pressure, with boiling point at 165 K and a narrow window (3.9 K) of liquid phase.

Table 1: Energy levels of atomic orbitals for isolated atoms of xenon measured via photoelectron spectroscopy [28]. The vacuum energy level is set to 0 eV.

|  | Energy [eV] |
|---|---|
| $5p_{\frac{3}{2}}$ | -12.1 |
| $5p_{\frac{1}{2}}$ | -13.4 |
| $5s$ | -23.3 |
| $4d_{\frac{5}{2}}$ | -67.5 |
| $4d_{\frac{3}{2}}$ | -69.5 |

Figure 1: Left: The face-centered-cubic crystal structure of solid xenon, with the primitive rhombohedral unit cell shaded in gold. The arrows indicate the primitive lattice vectors. The experimentally measured value of the cubic lattice parameter extrapolated to 0 K is 11.59 a.u. [33]. Right: Representative supercell of liquid xenon extracted from our Monte Carlo simulation (see details in Sec. 3.4) and used as input for liquid-phase DFT calculations. This region was randomly sampled from the generated atomic distribution and contains 30 atoms within a cubic cell with a lattice parameter of 24.88 a.u, yielding the measured atomic density inside the detector $\rho_{det} = 1.949 \times 10^{-3}$ atoms/a.u.$^3$ [34, 35].

### 3.1.1 Atom

The energy levels of the outer shell electrons in isolated atomic xenon have been measured using photoelectron spectroscopy [28] and are reported relative to the vacuum level in Tab. 1. Note the splitting of the $5p$ and $4d$ levels due to spin-orbit coupling.

### 3.1.2 Liquid

The measured density in the liquid phase at 183.18 K with an overpressure of 2.53 bar is $2.02 \times 10^{-3}$ atoms/a.u.$^3$, with peaks in the atomic radial distribution function (Fig. 5) at $\sim 8.5$ a.u. and $\sim 14.7$ a.u., corresponding to the first and second nearest neighbouring atoms, respectively [36]. In addition, the refractive index of liquid xenon has been measured at various wavelengths (Fig. 6) [37, 38] in Sec. 3.4. We will benchmark our calculations to these properties in Sec. 3.4.

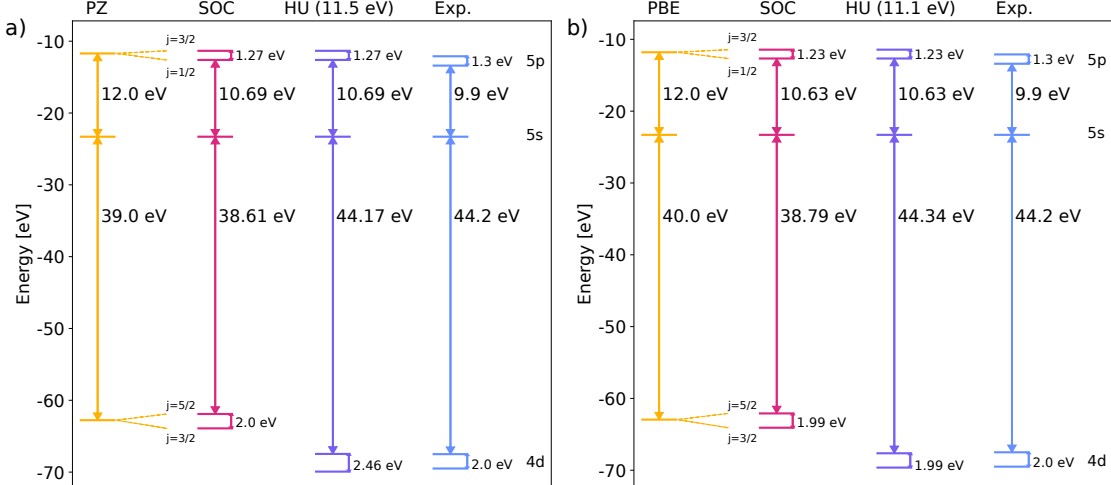

Figure 2: Valence electron energy levels of atomic xenon calculated using DFT, with the LDA PZ (a) and GGA PBE (b) XC functionals. Spin-orbit coupling is included in the second columns (labelled SOC), and both spin-orbit coupling and a Hubbard-$U$ on the Xe 4$d$ states in the third (labelled HU). The last column of each panel shows the experimentally measured values [28]. The calculated levels are aligned so that their 5$s$ orbitals have the same energy as the experimental value of -23.3 eV (see Tab. 1).

### 3.1.3 Solid

Xenon crystallizes at 161.1 K in the face-centered-cubic (fcc) structure (Fig. 1, left panel) with a zero-kelvin extrapolated value of the cubic lattice parameter of 11.59 a.u. [33], corresponding to a density of $2.57 \times 10^{-3}$ atoms/a.u.$^3$ and a nearest neighbour distance of 8.20 a.u. Solid Xe is an insulator with reflection spectra and photoemission measurements indicating a band gap of about 9.3 eV at the $\Gamma$ point [39,40] that decreases by $\sim 0.2$ eV with increasing temperature up to the melting point due to lattice expansion [39]. Photoemission experiments indicate that the width of the highest occupied valence band, derived from the 5$p$ atomic orbitals, is approximately 3 eV [40].

### 3.2 Density-functional calculations for atomic xenon

We begin by calculating the properties of isolated xenon atoms, with the goal of identifying a DFT setup that correctly reproduces the experimentally measured electronic energy levels of the valence electrons. Since `Quantum Espresso` employs periodic boundary conditions, we place a single Xe atom in an otherwise empty cubic box, and increase the size of the box until further change in the calculated total energy is less than 1 $\mu$eV; this was achieved at a cubic unit cell length of $\sim 27$ a.u. For more detailed information about our computational scheme and parameters, see Appendix A.

We test two different approaches for modeling the XC functional: the local density approximation (LDA) and the generalized-gradient approximation (GGA), in their Perdew-Zunger (PZ) [24] and Perdew-Burke-Ehrenhof (PBE) [41] implementations, respectively. In Fig. 2, we show our valence electron energy levels calculated using PZ LDA (left) and PBE GGA (right), together with the experimentally measured values in the last column of each panel, with the energies of the 5$s$ states aligned for comparison.[5] We see that the two functionals in their basic

---

[5]Note that our ground-state DFT calculations do not yield information about the ionisation energy, which is the energy difference between the highest occupied 5$p$ level and the vacuum level.

form (first two columns) yield similar values, with the correct ordering of the energy levels in both cases although with quantitative differences from the measured values. In both cases, however, the observed splittings of the 5*p* and 4*d* states are lacking and the position of the 4*d* levels is ∼ 5 eV too high, so that the energy difference between the 5*s* and 4*d* levels is too small compared to experiment. The introduction of spin-orbit coupling (SOC, second columns from left) corrects the first problem, with the 5*p* and 4*d* levels splitting according to their different *j* values and with splitting values in excellent agreement with experiment for both functionals. The underestimated binding energy of the 4*d* orbitals is largely a result of the fact that, in standard DFT calculations, each electron incorrectly interacts with the Coulomb and exchange-correlation potential generated by its own charge, artificially increasing its energy, particularly in localized orbitals. We correct for this self-interaction error by introducing a Hubbard-*U* correction on the 4*d* states following the rotationally invariant approach of Dudarev et al. [42] and varying the value of the effective Hubbard parameter, $U_{\text{eff}}$, until the experimental 5*s*-4*d* energy difference is correctly reproduced.[6] We find that $U_{\text{eff}}$ values of 11.5 eV (LDA) and 11.1 eV (GGA) bring the 5*s*-4*d* energy difference into good agreement with experiment (HU, third columns from left), although at the expense of increasing the spin-orbit splitting of the 4*d* states to slightly more than the experimental value.

In summary, our analysis indicates that both LDA and GGA functionals, with inclusion of SOC and appropriate adjustments to correct for self-interaction, are able to reproduce the measured energy levels of atomic Xe. In the next section, we assess which of these functionals provides the best description of the crystalline solid phase.

## 3.3 Density-functional calculations for solid xenon

### 3.3.1 Van der Waals correction

As mentioned above, since xenon is a noble gas with filled electron shells, the van der Waals interaction is key to the formation of the condensed phase. However, because of its intrinsic non-local character, the van der Waals interaction is not captured in either local or semi-local functionals, such as the PZ and the PBE functionals we used above. Fortunately, a number of corrections to standard XC functionals have been developed to specifically treat van der Waals interactions and several are implemented within the `Quantum Espresso` package. We tested six different forms of van der Waals correction – vdw-DF, vdw-DF2, rVV10, DFT-D, DFT-D3 and mbd_vdw – the physics of which are described in Appendix B, by calculating the lattice parameters that they yield for solid crystalline xenon. Our calculated lattice parameters are shown in Fig. 3, obtained as corrections to the LDA PZ (left panel) and GGA PBE (right panel) XC functionals. Computational details are provided in Appendix A. The colours denote the progressive inclusion of the Hubbard-*U* correction (central purple bars) and of SOC (right pink bars) in our calculations. We find that the values of the lattice parameter are approximately independent of both SOC and the Hubbard-*U* correction. We also see that the PZ functional yields lattice parameters which are too small, a typical problem of the LDA approximation. PBE performs better for this property, especially when van der Waals corrections introduce the missing attractive contribution. We note that the three non-local functionals vdw-DF, vdw-DF2 and rVV10 are not implemented in combination with PZ in the `Quantum Espresso` code.

Based on this analysis, we find two functionals, the rVV10 and DFT-D corrections to PBE, that closely reproduce the experimental value of the zero-kelvin-extrapolated lattice parameter of 11.59 a.u. [33]. Since rVV10 cannot be used together with spin-orbit coupling in `Quantum`

---

[6]While the DFT + Hubbard-*U* method was developed to incorporate an explicit Coulomb repulsion between electrons in a partially filled orbital manifold on the same atom, since the effect of the $U_{\text{eff}}$ parameter is to lower the energy of occupied states it can also be used to correct for self-interaction in filled orbitals. In this case, the value of the parameter $U_{\text{eff}}$ has the effect of shifting the selected filled orbital down in energy by $\sim \frac{U_{\text{eff}}}{2}$.

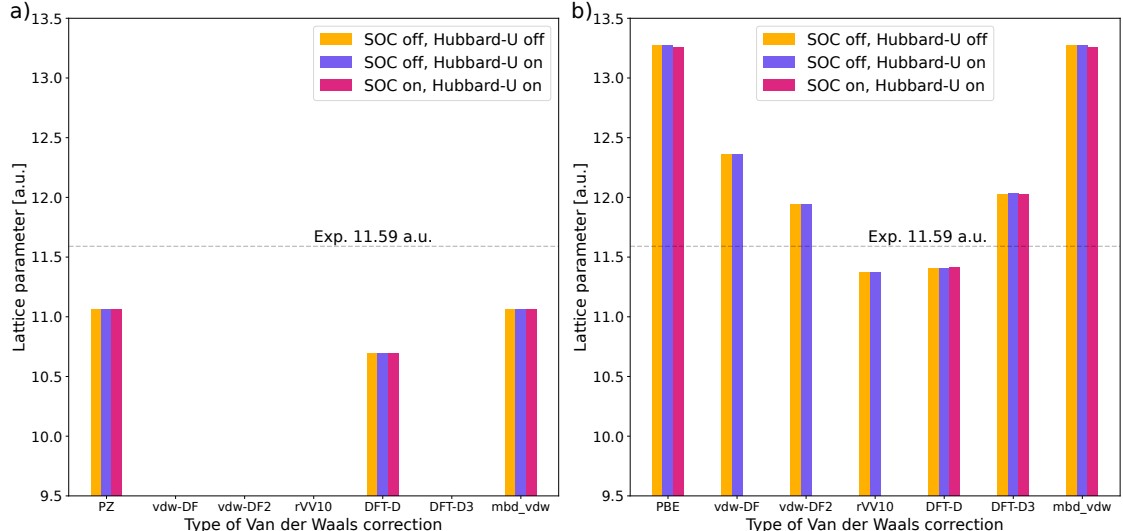

Figure 3: Calculated values of the lattice parameter for crystalline xenon using several van der Waals corrections to the PZ (a) and PBE (b) XC functionals. Yellow bars denote calculations without Hubbard-$U$ and SOC correction, purple bars include the Hubbard-$U$ and do not include SOC correction, while calculations marked in pink include both corrections. vdw-DF, vdw-DF2, rVV10 and DFT-D3 corrections are not implemented with the PZ XC functional, while the vdw-DF, vdw-DF2 and rVV10 implementations do not allow inclusion of SOC. The horizontal dashed lines indicate the experimental value of the lattice parameter [33].

`Espresso`, we select the DFT-D correction to the PBE XC functional (with calculated lattice parameter of 11.42 a.u.) as our choice to treat van der Waals interactions and use it in the subsequent calculations.

### 3.3.2 Calculated electronic properties

Fig. 4 shows our calculated band structure and density of states (DOS) for solid crystalline xenon, calculated with (pink) and without (blue) spin-orbit coupling, with an 11.1 eV Hubbard-$U$ correction on the $4d$ states and using the DFT-D van der Waals correction to the PBE XC functional. The top of the valence band is set to 0 eV.

The highest occupied valence bands are dominated by the three $5p$ atomic orbitals. We see that the spin-orbit coupling lifts degeneracies in the $5p$ bands, causing, for example, a 1.42 eV splitting at the $\Gamma$ point ($\mathbf{K}=0$) of the band originating from the atomic $j=\frac{1}{2}$ manifold, which is slightly larger than the 1.23 eV splitting in the atom. The weak van der Waals bonding causes each sub-band to broaden by $\sim 1$ eV, leading to a total bandwidth for the $5p$ manifold of $\sim 2.5$ eV, consistent with photoemission data [40]. The calculated band gap of 5.81 eV shows the usual DFT underestimation compared to the experimental value (9.3 eV [39,40]). We note that this does not affect our calculations of DM-induced ionisation rates, where we position the occupied levels with respect to the vacuum level by using the measured ionisation potential. The $5s$ and $4d$ bands (not shown) are largely unaffected by the crystal environment; they remain narrow and occur at roughly the same energies as in the isolated atom. We also notice that the first empty conduction bands show a broad dispersion with parabolic shape at gamma, indicating that a plane-wave state of a free electron might already approximate well these states.

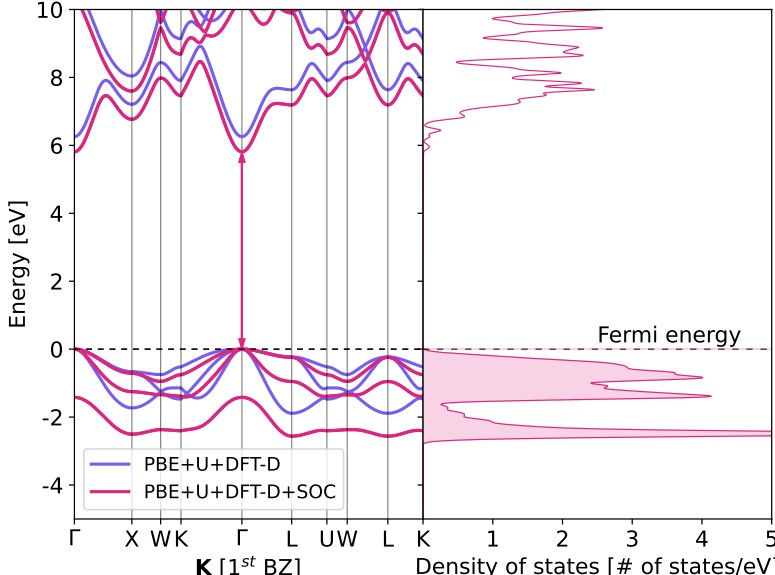

Figure 4: Calculated band structure (left) and density of states (right) for solid xenon with (pink) and without (blue) SOC effects. The pink arrow indicates the band gap with SOC. The introduction of SOC lifts the degeneracy in the $5p$-valence bands.

## 3.4 Density-functional calculations for liquid xenon

In order to use DFT (which is by construction a 0 K technique) to simulate the liquid phase of xenon, which occurs only at finite temperature, we adopt a hybrid approach, combining a classical Monte Carlo (MC) simulation of the atomic distribution at the experimental temperature followed by a 0 K calculation of the electronic structure for this atomic distribution performed with DFT.

For our MC simulation, we use a Lennard-Jones potential with parameters $\varepsilon = 0.02$ eV and $\sigma = 7.45$ a.u. [43] and the usual analytical form:

$$V(r) = 4\varepsilon \left[ \left( \frac{\sigma}{r} \right)^{12} - \left( \frac{\sigma}{r} \right)^{6} \right]. \tag{31}$$

Following the usual MC scheme, atoms are randomly selected and then displaced by a random distance up to a maximum value of $\sim 1.89$ a.u. (1 Å) in any direction, with the steps accepted or rejected according to the Metropolis-Hastings algorithm [44, 45], until the system's free energy is minimized. We set the number of trial steps per atom to 500 (with consistency checks at 1000 and 1500). The algorithm also requires a distance cutoff for the interatomic potential, above which the potential is set to zero: this was set to be half the length of the simulation box.

Our cubic simulation box contains 13500 atoms and we set the edge lengths to $\sim 189$ a.u (100 Å) to impose the experimental density of liquid xenon used in the detectors, $\rho_{det} = 1.949 \times 10^{-3}$ atoms/a.u.$^3$ [34, 35]. We simulate the detector temperature of 177 K; note that this temperature is higher than the boiling point of xenon at standard conditions due to an overpressure of 1.94 bar adopted in liquid xenon detectors [46]).

Fig. 5 shows the calculated radial distribution function (RDF) $g(r)$ with these parameters (solid pink line). We find satisfactory agreement with experimental RDF values (yellow dots) measured at 183.18 K and a density of $\rho_{exp} = 2.02 \times 10^{-3}$ atoms/a.u.$^3$ ($\sim 3\%$ higher than our simulation) [47]. (We note that more modern diffraction studies [48, 49] do not show the small peak at $\sim 11.5$ a.u.; we chose to include these data since the reported pressure

and temperature are closest to our values.) As further validation, we run an additional MC simulation at temperature near absolute zero and at the experimental density of solid xenon of $\rho_{solid} = 2.586 \times 10^{-3}$ atoms/a.u.$^3$ [33]. We obtain an fcc structure and sharp Dirac-delta-like peaks in the RDF as shown in solid blue lines in Fig. 5. The MC-simulated inter-atomic spacings are slightly larger than the calculated DFT values (blue dashed lines) due to the larger experimental lattice parameter used here.

Next, we extract multiple computationally tractable 30-atom cubic "supercells" from the results of the MC simulation for liquid xenon. These are then used, with periodic boundary conditions, as input to our DFT and DM-electron scattering rate calculations. The supercells are obtained by randomly sampling cubic blocks of edge length 24.88 a.u. from the MC-calculated atomic distribution; this density matches the experimental value of the DM detector $\rho_{det}$. We exclude supercells that result in atomic spacings less than 6.6 a.u. when periodic boundary conditions are applied. An example of a representative supercell is shown in the right panel of Fig. 1, and the average RDF calculated for the supercells generated using this procedure is shown as the dashed pink line in Fig. 5. More details on the supercell construction procedure are given in Appendix A.

Finally, we use DFT to compute the electronic properties of our liquid xenon supercells (see again Appendix A for computational details). As a first test of the validity of the method, we compute the refractive index (for more details, see Appendix C), which is shown in Fig. 6 along with the available experimental data points [37, 38]. Importantly, our DFT results reproduce the refractive index at $\lambda = 178$ nm, which is the wavelength of liquid xenon's own scintillation light (transparency of liquid xenon to its own scintillation light is one of the advantages of using this material as a detector medium [46]). Our setup also gives reasonable agreement for the long-wavelength limit.

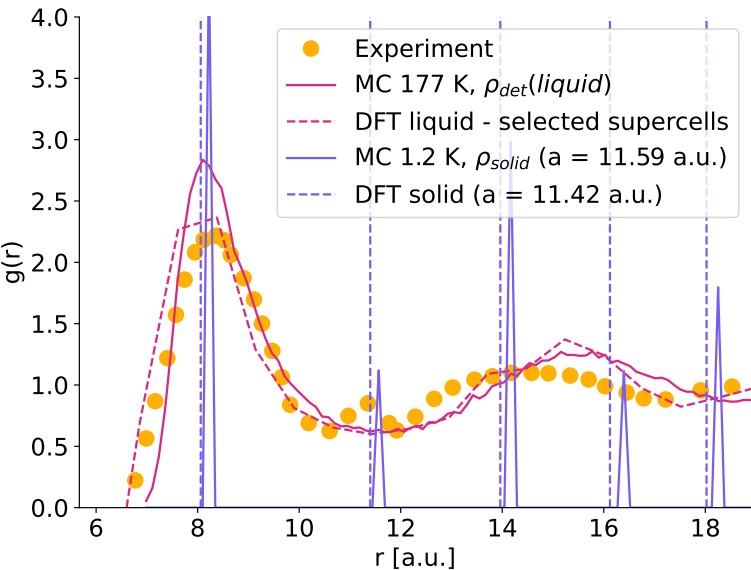

Figure 5: Radial distribution function for MC-simulated liquid xenon at 177 K with Lennard-Jones potential (pink solid line) compared to the experimentally measured distribution at 183 K (yellow dots) [47]. The pink dashed line shows the reconstructed distribution averaged over the supercells that we selected for the calculation of DM-electron scattering rates. The blue peaks are the results of the MC simulation of solid xenon near absolute zero with lattice parameter $a = 11.59$ a.u. and the blue dashed lines show the interatomic distances at the calculated DFT lattice parameter $a = 11.42$ a.u.

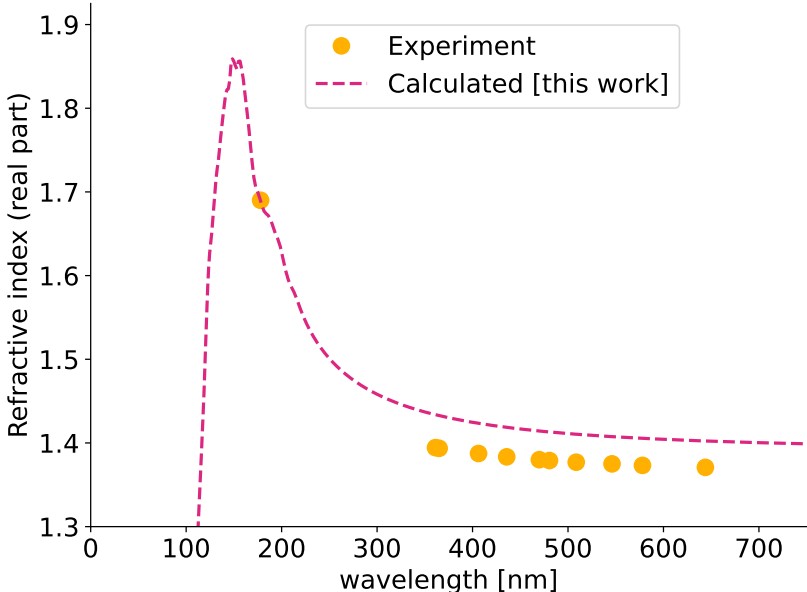

Figure 6: Calculated (pink dashed line) and measured (yellow dots) [37, 38] refractive index of liquid xenon. The data point at wavelength $\lambda = 178$ nm (the scintillations light wavelength) was measured at 170 K [37] and the points at higher wavelengths were measured at 178 K [38].

Next, we compare the calculated DOS of the liquid phase with those of the crystalline solid and the atom. Fig. 7 shows the calculated DOS for liquid xenon averaged over five representative supercells constructed as described above (pink), with those of the atom (orange vertical lines) and the solid (blue) for comparison. The levels are positioned so that the low-lying narrow 4$d$ states (panel a)) have the same energy in all phases and the vacuum level is set from the ionization potential of the isolated atom. In the zoom of panel b) we can see that, similarly to the solid case, the only orbitals that broaden appreciably into a band are the 5$p$, with a bandwidth very close to that of the solid ($\sim 3$ eV). As a result, the highest occupied level is now closer ($-11.05$ eV, so that $\Phi = +11.05$ eV) to the vacuum level than in the isolated atom (where $\Phi = 12.1$ eV). We reiterate that the ionisation potential, $\Phi$, which is the energy difference between the highest occupied level and the vacuum, is smaller in the condensed phase than the work function $W$, since the latter includes the energy cost for the electron to escape the surface. This will be important later in our analysis of exclusion limits.

To further understand the role of bond formation in the condensed phase, we plot in Fig. 8 representations of the real-space charge density for the solid (left) and liquid (right). For the solid, we show a contour plot of the calculated valence charge density in the (111) plane in the front panel. It is clear that the charge density is highly spherical around the atomic sites, with small deviations from spherical symmetry visible only at small electron densities below $\sim 0.002\, e$/a.u.$^3$ The yellow blobs within the crystal structures indicate the isosurfaces of charge density at $0.015\, e$/ a.u.$^3$ (panel a)) and $0.001\, e$/a.u.$^3$ (panel b)). The first value is chosen because it is the value of the charge density at the core radius of the pseudopotential for the 5$p$ electrons, $r_c^{5p} = 2.99$ a.u., below which the wave functions are smoothened due to the pseudopotential approximation. This point will be discussed further in Sec. 3.5. We see that at this value of charge density, the charge distribution is effectively spherical and centered around the atoms. Panel b) emphasizes the non-spherical bonding charge between the atoms, which is only apparent at low density (isosurface $0.001\, e$/a.u.$^3$).

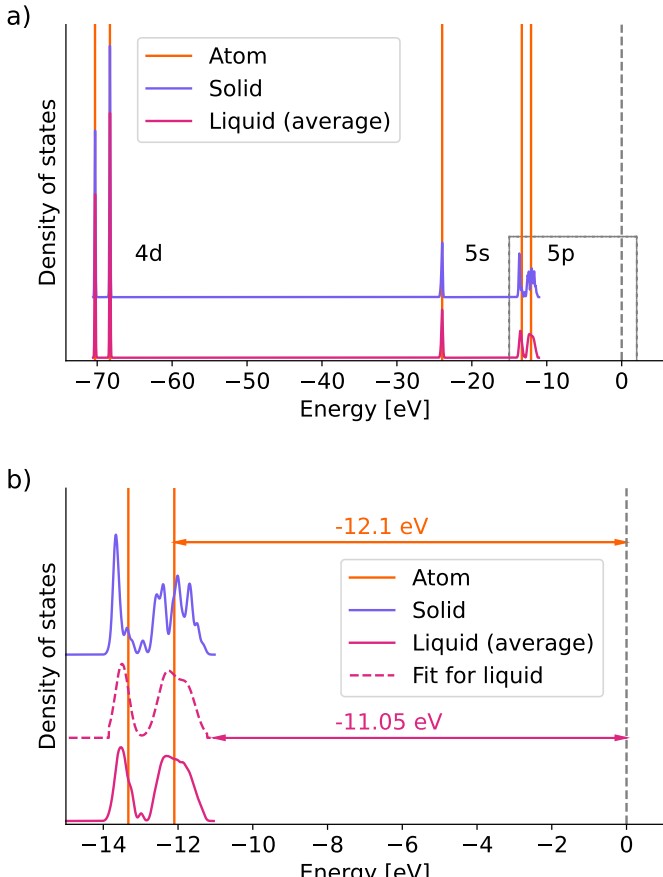

Figure 7: a) Computed valence band density of states of liquid xenon averaged over supercells (pink), solid crystalline xenon (blue) and isolated Xe atom (orange). The 5$p$ levels of the isolated atom are positioned according to the experimental ionisation energy of $-12.1$ eV, with zero (gray dashed line) indicating the vacuum energy [28]. The solid and liquid curves are positioned so that the 4$d$ levels match those of the isolated atom. b) Computed 5$p$ atomic, liquid and solid densities of states (gray box in panel a)) with the ionisation potentials for the atomic (orange) and liquid (pink) phases indicated. The pink dashed line is the best fit of Eq. (32) to our calculated liquid density of states.

The right-hand side of Fig. 8 shows a contour plot of the charge density averaged perpendicular to the plane of a selected supercell of liquid xenon and compares the total charge density (panel c), top) with its deviation from that of non-interacting atoms at the atomic sites (panel d), bottom). Note that the scale in panel d) is three orders of magnitude smaller than that of panel c), indicating that the charge participating in the chemical bonding is correspondingly smaller than the overall charge, consistent with the weak van der Waals interactions.

## 3.5 Pseudopotentials

Before concluding our discussion on the DFT treatment of xenon, we make some remarks about the pseudopotential approximation in the context of calculations of DM-electron scattering rates. The pseudopotential method replaces the central Coulomb attraction of the valence electrons to the positive ion composed of the nucleus and the tightly bound core electrons with a smoother effective potential, the pseudopotential. The pseudopotential is constructed so that

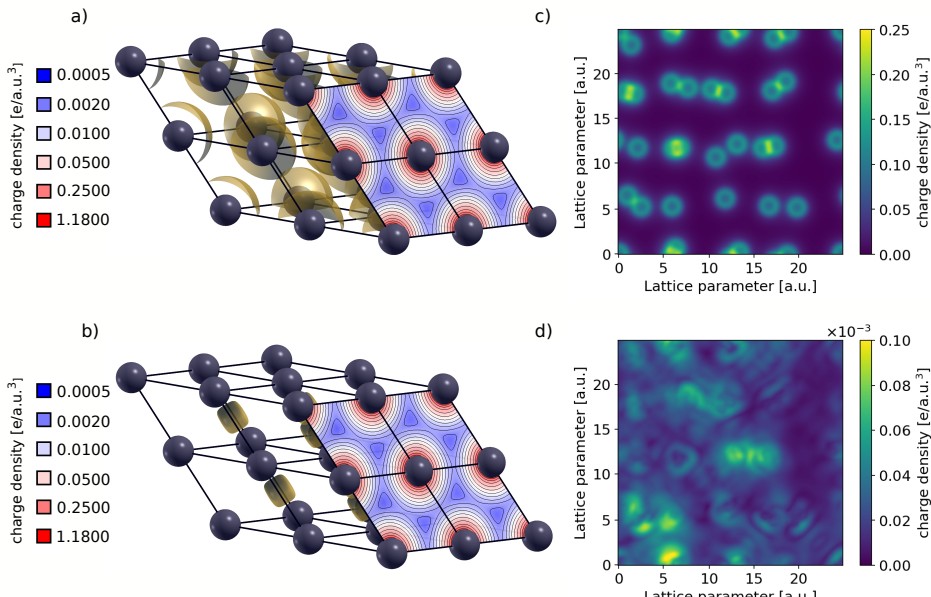

Figure 8: Computed electron density in real space for solid and liquid xenon with PBE functional, DFT-D vdW correction and $U_{\text{eff}} = 11.1\,\text{eV}$. Panels a) and b) show contour plots of the charge density of solid xenon in the (111) plane (front surfaces) combined with charge density isosurfaces (gold blobs) at the values of $0.015\,e/\text{a.u.}^3$ (a) and $0.001\,e/\text{a.u.}^3$ (b). The value of $0.015\,e/\text{a.u.}^3$ corresponds to the charge density value at the pseudopotential core radius $r_c^{5p} = 2.99\,\text{a.u.}$ for $5p$ electrons. Panels c) and d) show the in-plane average of the charge density of liquid xenon for one of our selected supercells. In panel c), the total valence charge density is shown; in panel d), the charge density of isolated xenon atoms placed at the atomic sites was subtracted and the absolute value of the charge density difference is shown.

it and the pseudowavefunction obtained from solving the corresponding Schrödinger equation are identical to the true potential and wave function outside of some core radius $r_c$, with the requirement that the true and pseudo eigenenergies are identical. The pseudowavefunctions obtained from this construction have fewer nodes than the true wave functions, and as a result, their expansion in a plane-wave basis is computationally tractable. In Fig. 9 we plot the radial distribution of the charge from the Xe $5p$ electrons calculated in the Roothan-Hartree-Fock (RHF) approximation, which is the standard literature choice for modeling atomic Xe for DM-electron scattering calculations [10,16,21], and indicate the core radius of the pseudopotential used in this work, $r_c^{5p}$, with the vertical dashed line. We see the strongly oscillating behaviour of the RHF radial charge density in the region within the core radius of the pseudopotential, and that most of the $5p$ electron charge lies within the core region.

The pseudopotential approximation is widely used in the Materials Physics community and is well established to yield results comparable to calculations that treat the potentials of all the electrons explicitly (so-called all-electron methods) for standard condensed matter properties. The excitation of electrons by light DM can, however, involve a large momentum transfer $\mathbf{q}$, up to the order of $\sim 100\,\text{keV}$. As previous work has pointed out [21, 51, 52], this amount of momentum is sufficient to probe short length scales comparable to the wavelengths of the fast oscillations of the wave function in the core region. This means that high momentum transfer scattering rates are artificially suppressed when a smoothened pseudowavefunction is used for the valence electrons.

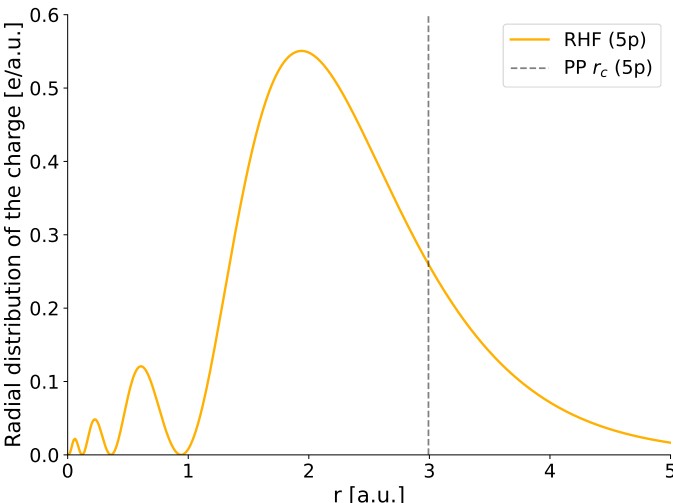

Figure 9: Radial distribution of the charge in the $5p$ RHF atomic orbital of xenon, calculated using the coefficients tabulated in ref. [50]. The core radius of the pseudopotential, $r_c^{5p} = 2.99$ a.u., used in this work is shown with the vertical dashed line.

One solution to this problem was shown in the work of Griffin et al. [21], where a so-called all-electron reconstruction of the DFT-obtained pseudowavefunctions was performed within the projected-augmented-wave (PAW) method [53] to compute scattering rates in crystalline Si and Ge targets. This method reconstructs the all-electron wave functions by means of a linear transformation connecting pseudo and all-electron atomic basis functions and the use of specific projector functions [21, 53]. Such a reconstruction is also used for example in the calculation of chemical shifts in nuclear magnetic resonance spectra, where the details of the valence electron wave functions in the region of the atomic nuclei are important. Another possibility is to avoid the use of pseudopotentials completely by using an all-electron code, although this would be prohibitively computationally expensive for the large numbers of atoms and electrons per atom considered here. We will show in the next section that the weak nature of the van der Waals interactions in condensed xenon allows for an even simpler work-around to this problem in this case.

## 3.6 Analytical fitting of the density of states

Finally, we perform an analytical fitting of the DOS of the liquid phase that we will use later to compute DM-induced ionisation rates (see the next Sec. 4). We find that the DOS of the $5p$ electrons of the liquid phase can be approximated as follows

$$\text{DOS}(E_e) = \frac{\Theta(E_e - t_1)\Theta(t_2 - E_e)}{N_2 - N_1} \sum_{i=1}^{3} a_i e^{-\frac{(E_e - \mu_i)^2}{\sigma_i^2}}, \tag{32}$$

where the energy $E_e$ was defined below Eq. (18) such that $E_e = 0$ corresponds to an energy of $-\Phi$ relative to the vacuum, and

$$N_j = \frac{\sqrt{\pi}}{2} \sum_{i=1}^{3} a_i \sigma_i \text{erf}\left(\frac{t_j - \mu_i}{\sigma_i}\right), \tag{33}$$

ensures that the DOS of the $5p$ electrons is normalized to six. The best-fit energy cutoffs are $t_1 = -2.82000$ eV and $t_2 = -0.18633$ eV, and the Gaussians are characterised by the parameters in Tab. 2. The resulting best-fit $\text{DOS}(E_e)$ is plotted as the pink dashed line in Fig. 7 b).

Table 2: Best-fit parameters for the three Gaussian terms in Eq. (32) approximating the DOS of the 5$p$ electrons in the liquid phase.

| $a_i$ | $\mu_i$ [eV] | $\sigma_i$ [eV] |
|---|---|---|
| 1 | $-1.27468$ | 0.33845 |
| 0.81422 | $-0.73646$ | 0.34622 |
| 1.19924 | $-2.46349$ | 0.25364 |

We find that the DOSs of the remaining electron shells (5$s$ and 4$d$) are well approximated by delta functions centered at the DFT-predicted orbital energies.

# 4 Calculation of the dark matter-induced ionisation rate in liquid xenon

## 4.1 Overture: General observations

By combining the DFT results from Sec. 3 with the gradient expansion of the ionisation rate in Eq. (19), we now develop a computational framework to calculate the expected rate of DM-induced ionisation events in liquid xenon detectors. Our framework builds on three observations. We list them here, and elaborate on them in the next subsections 4.2 and 4.3:

1. The electron binding energies / densities of states of the highest occupied 5$p$ levels are strongly dependent on the xenon phase, as shown in Fig. 7. In our calculation of the scattering rate and exclusion limit plots, therefore, we will explicitly determine the influence of the DOS of the liquid phase, in comparison to the isolated energy levels of the atom.

2. As shown in Eq. (14), when the Fourier transform of the final-state electron wave function peaks at a definite value of the linear momentum (as should be the case in ionisation processes), the ionisation rate can be explicitly expressed in terms of the initial-state electron density $\rho_e^{n\ell}(\mathbf{k}' - \mathbf{q})$ and the derivatives $\nabla_{\mathbf{k}'}\rho_e^{n\ell}(\mathbf{k}' - \mathbf{q})$ and $\nabla_{k_i'}\nabla_{k_i'}\rho_e^{n\ell}(\mathbf{k}' - \mathbf{q})$. We will see in section 4.2 that the density and its derivatives are approximately independent of the xenon phase. In particular, we note that the deviations in the 5$p$ densities between the atom and the condensed phases are several orders of magnitude smaller than those introduced by use of the pseudopotential. Therefore, to facilitate comparison with earlier calculations for atomic xenon, we will model the initial-state electron densities of the liquid using the isolated atom RHF wave functions, which in addition do not suffer from the pseudopotential approximation.

3. Concerning the final state, we showed in Sec. 2 that the gradient terms in the expansion of Eq. (14) are associated with deviations from a plane wave. We find these terms to be non-zero and approximately independent of the xenon phase. Therefore we proceed by using the standard literature choice of positive energy solutions of the hydrogen atom Schrödinger equation for the final state (see Sec. 2) enabling direct comparison of our results with previous work [12, 17, 54].

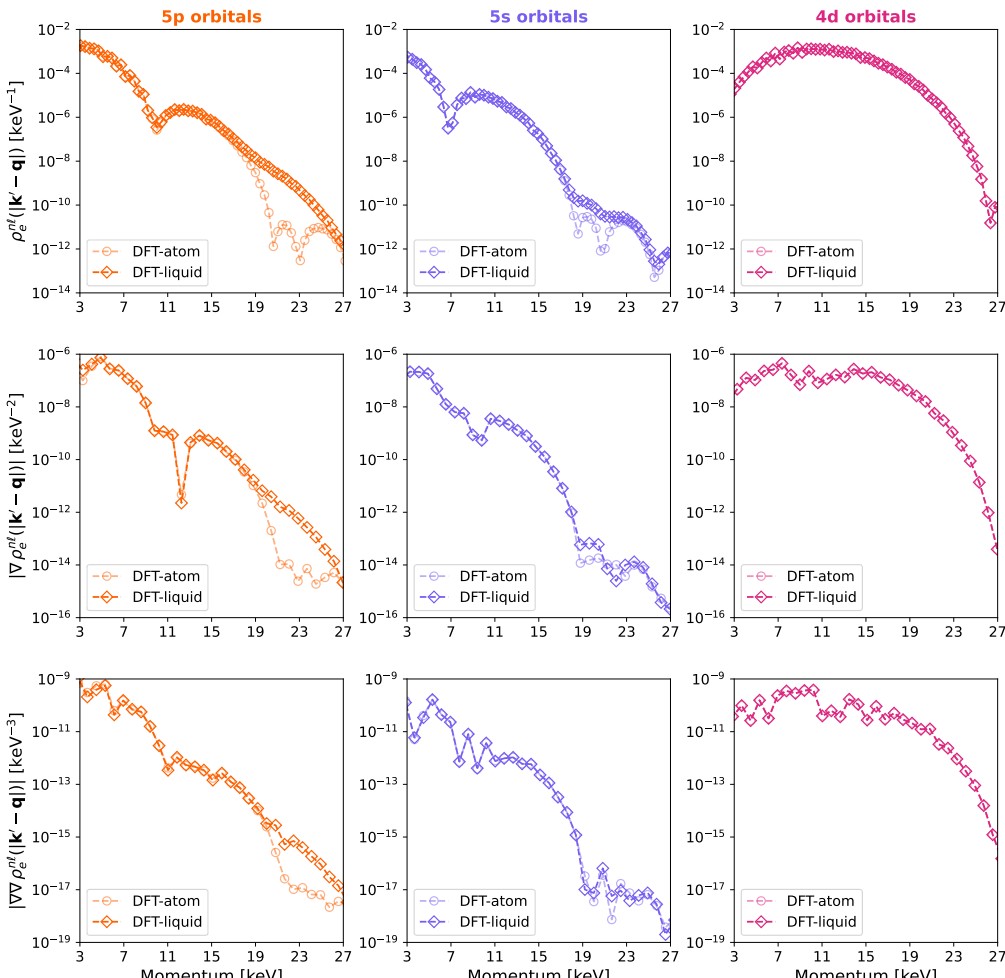

Figure 10: Orbital-resolved electron density $\rho_e^{n\ell}(|\mathbf{k}' - \mathbf{q}|)$ (angular average) and its first and second derivatives (absolute values) for atomic and liquid xenon, calculated with DFT with PBE functional, DFT-D vdW correction and $U_{\text{eff}} = 11.1$ eV. The columns show (left to right) the $5p$ (orange), $5s$ (blue) and $4d$ (pink) orbitals, while the rows show (top to bottom) the density and its first and second derivatives, respectively. Notice the different y-axis scales and units for the three rows.

## 4.2 Intermezzo: On the role of the initial-state electron density

Fig. 10 shows $\rho_e^{n\ell}(\mathbf{k}' - \mathbf{q})$ and its first and second derivative for atomic and liquid-phase xenon computed with DFT, using a modified version of our `QEdark-EFT` [32] code with PBE functional, DFT-D vdW correction and $U_{\text{eff}} = 11.1$ eV. We defined $\rho_e^{n\ell}(\mathbf{k}' - \mathbf{q})$ in Sec. 2.2 Eq. (23) to be the electron density in momentum space for the $n\ell$-th energy level and showed that it is an important quantity for computing the scattering rate.

Within our region of interest of momentum (up to 25 keV), the $4d$ orbitals (third column, pink lines) of the two phases have indistinguishable densities and density gradients, and the $5s$ orbitals (second column, blue lines) show only minimal differences, consistent with the atomic-like behavior of the $4d$ and $5s$ orbitals in the liquid phase. On the other hand, the liquid and atomic values for the $5p$ orbitals (first column, orange lines) start to differ at around 16 keV, with the liquid phase having up to approximately two orders of magnitude higher density and density gradients in this range. This is consistent with the change in real-space charge density associated with the bond formation.

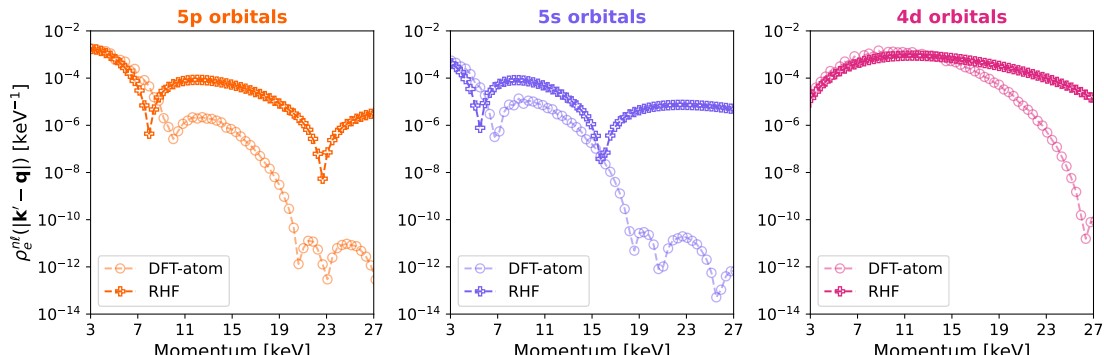

Figure 11: Orbital-resolved electron density $\rho_e^{n\ell}(|\mathbf{k}' - \mathbf{q}|)$ (angular average) of isolated xenon atoms calculated with DFT with PBE functional, DFT-D vdW correction and $U_{\text{eff}} = 11.1\,\text{eV}$, and RHF wave functions. The columns show (left to right) the 5p (orange), 5s (blue), and 4d (pink) orbitals.

In Fig. 11, we show again the orbital charge densities calculated for the isolated atom using DFT and compare them with those calculated using the RHF states. In Sec. 3.5, we discussed the pseudopotential approximation and mentioned its inability to reproduce high momentum components of the wave function in the core region. The comparison with the RHF density confirms this point, with all RHF orbitals having significantly higher density than the pseudopotential DFT orbitals in the high momentum regions. In particular, we notice that above 16 keV, where the phase of xenon plays a role in the DFT results, the RHF densities of all orbitals are orders of magnitude higher than the DFT. This leads us to the second observation that we made in the previous section: In the region of momentum space for which the atomic and the liquid phase yield different density and density gradients, the errors introduced by the pseudopotential approximation dominate over the difference between the atomic and liquid charge densities. We conclude therefore that we can best model the electron density $\rho_e^{n\ell}(\mathbf{k}'-\mathbf{q})$ and its gradients by using semi-analytical RHF states for isolated atoms. In addition, since the density of the 4$d$ electrons is completely phase independent at all momenta, we extrapolate that this phase independence also holds for the more tightly bound 4$p$ and 4$s$ orbitals. We therefore include these states in the calculation of the scattering rate via their RHF expressions.

Next, we address the question of how to best model the final state of the electron. In Sec. 2, examples of a positive-energy solution of the hydrogen problem and a plane wave were presented. Previous work has made use of both of these options [10, 16, 20], as well as of conduction band states [15, 18], or even a mixture of them [21], taking into account the broad range of energy transfer in the DM-electron scattering process. For liquid Xe the choice is unclear: While the final measured signal inside the xenon chamber is ionized electrons, which (neglecting the ions' electrostatic potential) would be associated with plane-wave states of free electrons, the electrons are first drifted through the medium by the applied electric field, which is likely associated with conduction band occupancy, suggesting the mixed approach of Ref. [21] could be the best choice. Here, we choose to proceed by using positive-energy solutions of the hydrogen problem, which allows us to directly compare our results with previous literature [16], leaving a more realistic description of the final state for future work. We note, in particular, that the density gradient terms - associated with deviations from a plane-wave final state (Eq. (14)) - are approximately the same for atomic and liquid phase (second and third row in Fig. 10), with the difference in the region above 16 keV due to the bonding charge being small compared to the core contribution. Therefore, the error coming from the choice of a specific final state should not depend on whether the xenon target is modelled as a liquid or as isolated atoms.

## 4.3 Finale: Our best estimate for the ionisation rate in liquid xenon

Building on the observations made in Sec. 4.1, we now introduce a framework to calculate the rate of DM-induced ionisation events in liquid xenon detectors. We recall Eq. (8), which is the function $\Delta(\mathbf{q}, \mathbf{v})$ that determines the ionisation rate $\mathscr{R}$ for the case where $\psi_1$ is a RHF wave function and $\psi_2$ is a positive-energy solution of the hydrogen problem, and replace the individual energy levels $E_{nl}$ by the DOS for electrons with quantum numbers $n$ and $l$,

$$\varrho^{n\ell}(E_e) \equiv \sum_{\hat{n}\hat{\ell}} \delta(E_e - E_{\hat{n}\hat{\ell}}) \, \delta_{\hat{n}n} \delta_{\hat{\ell}\ell} \,. \tag{34}$$

This yields

$$\Delta(\mathbf{q}, \mathbf{v}) = \frac{\pi}{2} \sum_{n\ell} \int \mathrm{d}E_e \, \varrho^{n\ell}(E_e) \int \frac{\mathrm{d}k'}{k'} \, W^{n\ell}(k', q) \delta\left(\frac{k'^2}{2m_e} + \Phi - E_e + \Delta E_\chi\right). \tag{35}$$

This equation coincides with Eq. (8), but explicitly includes the DOS $\varrho^{n\ell}(E_e)$.

Eq. (35) shows that the atomic to liquid xenon transition can affect the ionisation rate $\mathscr{R}$ in two ways: 1) by modifying $W^{n\ell}(k', q)$; 2) by changing the DOS $\varrho^{n\ell}(E_e)$ and the ionisation potential $\Phi$. Since $W^{n\ell}(k', q)$ can be expanded in gradients of the initial electron density, our results in Sec. 4.2 show that we can calculate $W^{n\ell}(k', q)$ using RHF atomic wave functions and the methods we developed in [16] and implemented in [55] with the DarkART package. In contrast, Fig. 7 shows that both the DOS $\varrho^{n\ell}(E_e)$ of the 5$p$ levels and the ionisation potential $\Phi$ are significantly affected by the atomic to liquid xenon transition. We also notice that, although the 4$d$ and 5$s$ states do not present any substantial broadening and have the same binding energy for the atom and the liquid phase, our DFT energies for these states were tuned to reproduce the experimental values and are different from the RHF values. Specifically, neglecting the splittings due to spin-orbit coupling, the DFT 4$d$ (5$s$) levels are 6.3 eV (1.74 eV) higher in energy than the RHF states.

Our best estimate for the ionisation rate $\mathscr{R}$ in liquid xenon thus relies on computing $W^{n\ell}(k', q)$ as in the standard treatment of DM-electron scattering in liquid xenon detectors, while replacing $\varrho^{n\ell}(E_e)$ and $\Phi$ with our DFT predictions. Specifically, for the 5$p$ electrons, we identify $\varrho^{n\ell}(E_e)$ with the fitting function in Eq. (32), for the 4$d$ and 5$s$ electrons we use delta functions at $E_{4d}^{\mathrm{DFT}} = -69.3$ eV and $E_{5s}^{\mathrm{DFT}} = -23.96$ eV and the ionisation potential $\Phi = 11.05$ eV.

Before applying the formalism described above to reassess the sensitivity of existing liquid xenon detectors, let us first briefly review the simplified detector response model often used in the literature to convert the *theoretical* ionisation rate $\mathscr{R}$, Eq. (4), into an *observable* rate of secondary scintillation events (the so-called $S2$ signal). Following [17], we assume that an ejected electron with kinetic energy $E_2 = k'^2/(2m_e)$ has 0 probability of being reabsorbed in the surrounding medium, while it generates a total number of charges $n_e$ (including the primary electron itself) with probability $P(n_e|E_2) = B(n_e - 1|n_1^0 + n_2^0, f_e)$. Here $n_1^0 = \mathrm{floor}(E_2/W)$ is the number of additional charges produced via direct ionisation, while $n_2^0$ is the number of additional charges produced via photoionisation associated with the de-excitation of individual atomic orbitals and is given in Tab. II of [10]. $W$ is the xenon work function which, as we mentioned above, is related to the ionisation potential $\Phi$, but contains the additional energy cost associated with the escape of the electron from the medium through its surface. Since we do not have access to this quantity within DFT, we allow it to vary from 12.1 eV to 16 eV, around the "central" value of 13.5 eV, which is compatible with [10,56]. Above, the function $B(a|b, c)$ denotes a binomial distribution with $a$ successes, $b$ trials and success rate $c$. For the success rate we assume $f_e = 0.83$ [57]. Finally, the $n_e$ charges produced in a DM-electron scattering event are assumed to be drifted to the detector surface, where they produce an observable number of photolectrons $S2$ with Gaussian probability of mean $g_2 n_e$ and variance $n_e \sigma_{S2}^2$.

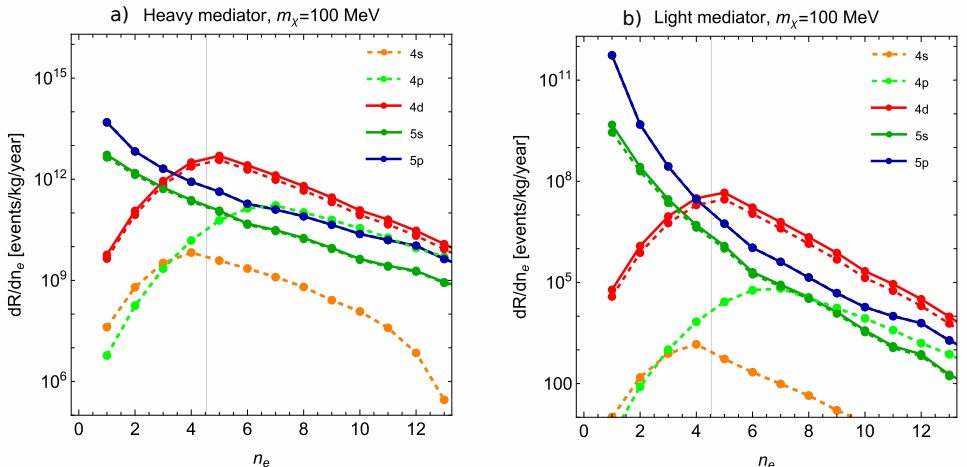

Figure 12: Orbital-resolved contribution to the expected differential ionisation rate as a function of the number of generated charges $n_e$ per interaction for a DM particle mass $m_\chi = 100$ MeV and for heavy (a) and light (b) mediator particles. Notice the different scale of the y-axis for the two panels. For the $4d$, $5s$ and $5p$ energy levels, dashed lines refer to the case of isolated RHF xenon atoms, whereas solid lines refer to our calculation scheme which uses DFT and includes the effect of the liquid phase. The sub-leading contributions from the $4s$ and $4p$ energy levels are calculated only for the RHF isolated atom. The vertical line shows the number of charges $n_e$ corresponding to a mean of 150 photoelectrons, which is the experimental threshold of XENON1T based on $\langle S2 \rangle = n_e g_2$. For illustrative purposes, in the figure we set the coupling constant $c_1 = 1$.

Within the detector response model above, the rate $d\Gamma^{n\ell}$ at which a number of photoelectrons between $S2$ and $S2 + dS2$ is produced in DM interactions with liquid xenon electrons corresponding to a $n\ell$ atomic orbital is given by

$$\frac{d\Gamma^{n\ell}}{dS2} = \varepsilon(S2) \sum_{n_e=1}^{\infty} \mathscr{G}(S2|n_e g_2, n_e \sigma_{S2}^2) \int dE_e \, P(n_e|E_2) \frac{d\mathscr{R}^{n\ell}(E_2)}{dE_2}, \qquad (36)$$

where $\mathscr{R}^{n\ell}$ is the contribution from the $n\ell$ orbital to the rate $\mathscr{R}$,[7] $\mathscr{G}(S2|n_e g_2, n_e \sigma_{S2}^2)$ is a Gaussian distribution of mean $n_e g_2$ and variance $n_e \sigma_{S2}^2$, and $\varepsilon(S2)$ is the detector efficiency.

By employing Eq. (36), we calculate the expected number of events in a sample of $S2$ bins, modeling the xenon target as a liquid. Here, we focus on two experiments for which it is straightforward to compare our predictions based on Eq. (36) with those obtained in the literature for the case of isolated xenon atoms: XENON10 [17, 54] and XENON1T [12]. For XENON10 (XENON1T), we use $g_2 = 27$ (33) and $\sigma_{S2} = 6.7$ (7) and model the corresponding detector efficiency and experimental exposure as summarised in [16]. In particular, we compare our theoretical predictions with the number of recorded $S2$ events reported in Tab. II of [16], assuming Poisson statistics for the number of events in each bin. Through this comparison, we obtain the 90% confidence level (C.L.) exclusion limits on the reference cross section

$$\sigma_e = \frac{c_1^2 \mu_{e\chi}^2}{16\pi m_e^2 m_\chi^2}, \qquad (37)$$

as a function of the DM mass. Here, $\mu_{e\chi}$ is the reduced DM-electron mass.

---

[7]It is obtained by replacing $\Delta(\mathbf{q}, \mathbf{v})$ with $\Delta^{n\ell}(\mathbf{q}, \mathbf{v})$ in Eq. (6).

SciPost Phys. **19**, 064 (2025)

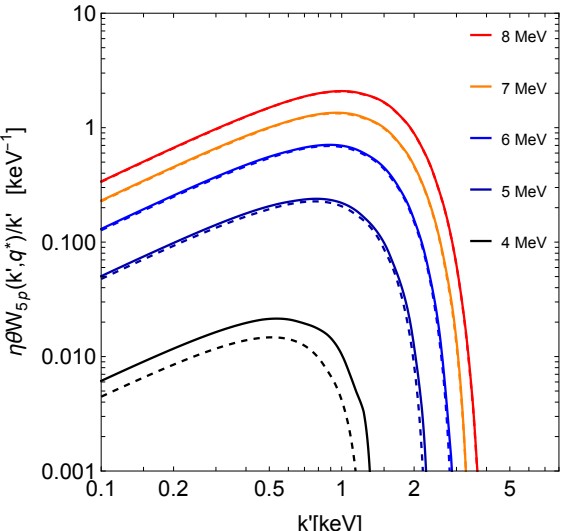

Figure 13: The integrand of the DM-electron ionisation rate (Eq. (38)) for the $5p$ states as a function of the electron final-state momentum $k'$, for DM masses between 4 and 8 MeV. Dashed lines are for atomic RHF xenon, solid lines for the liquid. For each value of $k'$, $q^*$ is set to the value of the momentum transfer $q$ that maximizes the integral.

We start the discussion of our results by first showing, in panel a) of Fig. 12, the contribution of the different atomic orbitals to the differential ionisation rate as a function of the total number of charges produced in an ionisation event for $m_\chi = 100$ MeV and a heavy mediator particle. Solid lines are calculated using our modeling of liquid xenon and the dashed lines using isolated RHF atoms. For the $4d$ states, there is a noticeable difference between the calculation for the RHF atom and our scheme, which is caused by the 6.3 eV energy difference between the $4d$ levels in the two calculation methods. The $5s$ states, which have a smaller energy difference of 1.74 eV, also show a smaller difference in the differential rate. Note that the $4s$ and $4p$ curves are shown only for the RHF case.

From Fig. 12, we can also see that the broadening of the $5p$ states and the resulting lower ionisation potential $\Phi$ associated with the liquid phase have no observable effect on the ionisation rate.

To better understand this unintuitive finding, we adopt an alternative form to express the contribution of the individual atomic orbitals to the differential ionisation rate, $d\mathcal{R}_{n\ell}$ (up to a multiplicative factor):

$$d\mathcal{R}_{n\ell} \propto q\,dq \int dk' f_{A/L}(k', q) \frac{W_{n\ell}(k', q)}{k'}, \tag{38}$$

where $W_{n\ell}$ is the atomic response function for the specific $n\ell$-th orbital and the functions $f_A$, for isolated RHF atoms, and $f_L$, for the liquid phase, are defined as:

$$
\begin{aligned}
f_A &= \eta\big(v_{min}(E_{n\ell}, k', q)\big)\theta\big(v_{esc} - v_{min}(E_{n\ell}, k', q)\big), \\
f_L &= \int dE_e\, \varrho^{n\ell}(E_e)\eta\big(v_{min}(E_e, k', q)\big)\theta\big(v_{esc} - v_{min}(E_e, k', q)\big),
\end{aligned}
\tag{39}
$$

where $\varrho^{n\ell}(E_e)$ is the unit-normalized density of states of the considered orbital. The velocity integral $\eta$ [18] and the minimum DM velocity $v_{min}$ are defined as

$$\eta(v_{min}) = \int \frac{\mathrm{d}^3 v}{v} f_\chi(\mathbf{v}) \theta(v - v_{min}) , \tag{40}$$

and

$$v_{min} = \frac{q}{2m_\chi} + \frac{\frac{k'^2}{2m_e} + \Phi - E_{n\ell}}{q} , \tag{41}$$

where $f_\chi(\mathbf{v})$ is the local DM velocity distribution in the detector rest frame and $\theta$ is a step function.[8]

In Fig. 13, we plot the the maximal contribution to the integrand of the ionization rate (Eq. (38)) for the $5p$ states as a function of the electron final-state momentum $k'$ by fixing the transferred momentum to the value $q^*$ that maximizes the contribution to the final integral. The colours indicate DM masses in the range [4, 8] MeV, whereas dashed and solid lines refer to isolated RHF atoms (calculated with the function $f_A$) and our liquid modeling (calculated with the function $f_L$), respectively. Since $q^*$ has been chosen as the value of $q$ that maximizes $f_A$ or $f_L$ for the given $k'$, the area beneath each line (which is proportional to the integral over $k'$) represents the largest contribution to the ionisation rate. While the results for the atom and liquid differ by roughly a factor of two for DM masses of 4 MeV, the difference reduces rapidly with increasing DM mass, and the two phases are indistinguishable for DM masses of 8 MeV. We conclude that the ionisation rate starts to be sensitive to the broadening of the $5p$ energy levels only for very low DM masses of $\sim 4$ meV.

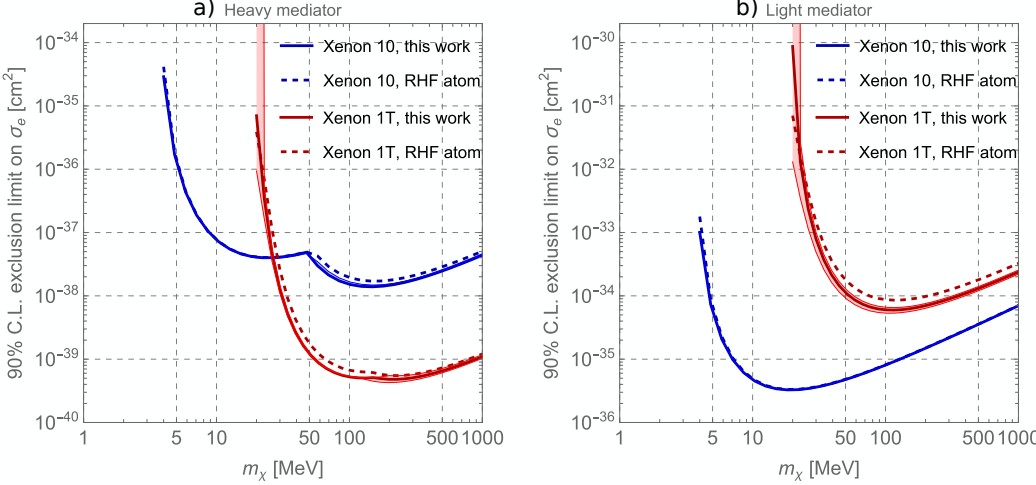

Figure 14: 90% C.L. exclusion limits on the reference cross section $\sigma_e$ as a function of the DM particle mass for XENON10 (blue) and XENON1T (red) and for heavy (a) and light (b) mediator particles. Notice the different scale of the y-axis for the two panels. Solid lines refer to our calculation framework which includes the effect of the liquid phase with $W = 13.5$ eV and the associated colored bands enclosing the 90% C.L. exclusion limits for W in the range 12.1 eV – 16 eV. Dashed lines correspond to the case of isolated RHF xenon atoms with $W = 13.5$ eV [16]. The change of slope of exclusion limits for heavy mediator in XENON10 is due to a division of the recorded events into two intervals depending on the recorded number of photoelectrons, which are kinematically accessible at different mass scales.

---

[8] The step function $\theta$ represents the kinematic requirements $v_\chi > v_{min}$ and $v_\chi < v_{esc}$.

Next, we show in panel a) of Fig. 14 our 90% C.L. exclusion limits on the reference cross section $\sigma_e$ as a function of the DM particle mass for XENON10 (blue) and XENON1T (red), assuming DM interactions mediated by a heavy mediator particle. Dashed lines correspond to isolated RHF xenon atoms [16], and solid lines to our computational framework for the liquid phase. In both cases $W = 13.5$ eV. The colored bands enclose the 90% C.L. exclusion limits when $W$ is varied from $12.1-16$ eV; this only affects the 90% C.L. exclusion limit when the number of additional charges produced via direct ionisation in an ionisation event, $n_1^0$, dominates over the number of additional charges produced via photoionisation associated with the de-excitation of individual atomic orbitals, $n_2^0$. We see that XENON1T is not sensitive to DM masses below $\sim 20$ MeV, which is a consequence of its signal threshold on the number of detectable photoelectrons (vertical line in Fig. 12). Its 90% C.L. exclusion limit is therefore insensitive to the broadening of the $5p$ states in the liquid phase, In contrast, the liquid-induced broadening of the $5p$ states starts to affect the exclusion limits of XENON10 in the region of DM masses close to 4 MeV, where we start to see a small reduction in the exclusion limit for the liquid compared to the atom. For both experiments, the differences at higher masses are a consequence of the different energies of the $5s$ and $4d$ states, with the empirical values used in our scheme leading to exclusion limits lower by a factor of up to $\sim 2$ than for the RHF atom.

Finally, we discuss the case of light mediator particles, shown in the b) panels of Figs. 12 and 14. We recall that this scenario has a factor proportional to $\frac{1}{q^2}$ in the formula for the ionisation rate, which increases the ionisation rate for lower momentum transfers and lower DM masses compared to the heavy mediator case, shown in the a) panels. In particular, the lower threshold on the number of detectable photoelectrons of XENON10 allows it to set more stringent limits in the light mediator scenario, where low-momentum-transfer events associated with a smaller number of photoelectrons have a higher weight. The differences between the RHF atom and our DFT liquid model are similar to those of the heavy mediator case, as expected.

## 5 Summary and outlook

In this work, we determined DFT parameters that provide a good description of xenon as an isolated atom and in its solid and liquid phases. We then used this set up to calculate the densities of states and the momentum-space electron density and its gradients for atomic and liquid xenon. Based on these quantities, we developed a computational scheme for computing DM-electron scattering rates, with the goal of critically analyzing the usual isolated-atom approximation.

Our DFT study highlighted several factors that are important for achieving an accurate description of liquid xenon. The spin-orbit coupling interaction must be included, particularly to obtain the correct energy broadening of the highest occupied $5p$ bands as well as the splitting of the more tightly bound $4d$ energy levels. The spurious self-interaction in the localized $4d$ states can be corrected by applying a Hubbard-$U$ correction, with a $U_{\text{eff}}$ parameter of 11.1 eV aligning the $4d$ energies with the measured photoemission values. For the condensed phases, van der Waals interactions must be included in the exchange-correlation functional, with the semi-empirical DFT-D correction to the PBE functional providing a good lattice parameter for the crystalline solid. Finally, a hybrid approach combining a classical Monte Carlo simulation of the atomic distribution and a subsequent DFT calculation of supercells sampled from the generated distribution is effective for modeling liquid xenon at low computational cost.

Comparing the electronic structures of the atom and the liquid phase, we found that, while the density of states of the $5p$ electrons for the liquid phase differs substantially from the discrete energy levels of the atom, the effect of condensation on the electron density is minimal compared to the error introduced by the pseudopotential approximation of our DFT calcula-

tions.

The calculation of DM-electron scattering rates concluded our work. We treated the leading spin-independent operator in the non-relativistic effective theory of DM-electron interactions [16], relevant for models such as the dark photon model where DM couples to the electron density [58], and considered interactions arising from the exchange of both a heavy and a light mediator particle. We developed a hybrid approach for calculating the DM-induced ionisation rate, in which the electron density is modeled by RHF wave functions while the DFT-calculated $5p$ density of states of the liquid phase and the DFT energies tuned to the photoemission values for the $4d$ and $5s$ states are used for the energy eigenvalues. Within this scheme, we computed differential ionisation rates and 90% C.L. exclusion limit curves for the XENON10 and XENON1T experiments and compared them with previous calculations for isolated RHF atoms [16]. We found an impact of up to a factor of $\sim 2$ on the exclusion limit curves in our calculations for the liquid phase compared to isolated atoms, depending on the DM mass, the experiment and the mediator scenario. Two distinct factors are important: The difference in calculated energy of the $4d$ and $5s$ states between the RHF atom and DFT, and the lower ionisation potential in the liquid phase, caused by the broadening of the $5p$ states. The latter reduces the cross section at low masses in the XENON10 experiment, due to its low signal threshold.

Our work provides a foundation for a more sophisticated treatment of sub-GeV DM-electron interactions in detectors based on liquefied noble gases, and we conclude by discussing the limitations of the approximations that have been taken. The choice of a positive-energy solution of the hydrogen problem as the final state, while widely used, might not be ideal because it omits the physics of the electron extraction to the detector's surface via the applied electric field. Here, recent developments in the angle-resolved photoemission spectroscopy (ARPES) community, where treatment of the electron final states as Bloch functions of Coulomb waves has recently been explored [59] is a promising starting point. The error introduced by the pseudopotential approximation could be addressed more rigorously via an all-electron reconstruction [21], allowing the use of DFT-obtained, rather than RHF, wavefunctions and preserving the effects of the liquid phase also at the level of the electron density. Finally, we expect different results for other DM-electron scattering models outside of the dark photon paradigm, which have been shown to give rise to additional material response functions [16, 18].

# Acknowledgments

The research presented in this paper made use of the following software packages, libraries, and tools: Matplotlib [60] NumPy [61], SciPy [62], WebPlotDigitizer [63], Wolfram Mathematica [64], XCrySDen [65] and VESTA [66]. DFT computations were performed on the Euler cluster provided by the ETH Zurich.

**Funding information**   R.C. acknowledges support from an individual research grant from the Swedish Research Council (Dnr. 2018-05029), and from the Knut and Alice Wallenberg Foundation via the project grant "Light Dark Matter" (Dnr. KAW 2019.0080). N.A.S., L.M. and M.M. were supported by the ETH Zurich, and by the European Research Council (ERC) under the European Union's Horizon 2020 research and innovation programme project HERO Grant Agreement No. 810451. R.C., N.A.S. and L.M. acknowledge funding also from the Swiss National Science Foundation under grant agreement no. 231539. M.M. was furthermore supported by the CTU Mobility Project MSCA-F-CZ-III under the number CZ.02.01.01/00/22_010/0008601, co-financed by the European Union.

**DATA**  The data associated with this project are available on the Zenodo platform, doi:10.5281/zenodo.14774056.

# A  DFT computational details

For atomic xenon, calculations were performed using only the Γ point of the Brillouin zone. For testing the different functionals, the energy cutoff for the plane-wave expansion $E_{\text{cutoff}}$ was set to 120 Ry. In the calculation of the electron density and its gradients (Fig. 10), we increased the cutoff up to 960 Ry to capture high-momentum-transfer DM-electron scattering processes to the extent possible within the pseudopotential approximation.

We tested two types of pseudopotentials: for the PZ XC functional, we used the ultrasoft (US) pseudopotential `Xe.pz-dn-rrkjus-psl.1.0.0.UPF` [67], whereas for PBE we chose the norm-conserving (NC) pseudopotential `Xe-ONCV-PBE-sr.upf` [68]. When spin-orbit coupling was introduced, their fully relativistic counterparts were used, namely `Xe.rel-pz-dn-rrkjus-psl.1.0.0.UPF` [67] and `Xe-ONCV-PBE-fr.upf` [68]. All these pseudopotentials treat the $4d$, $5s$ and $5p$ shells as valence electrons. The core radii for the pseudopotential that we selected, `Xe-ONCV-PBE-fr.upf` [68], and used for all calculations from Sec. 3.4 onward are $r_c^{4d} = 2.68$ a.u., $r_c^{5s} = 2.98$ a.u. and $r_c^{5p} = 2.99$ a.u. We also computed the properties of the isolated atom using a projected-augmented-wave (PAW) pseudopotential with the PBE XC functional and obtained the same energy levels as for the NC PBE pseudopotential.

For the crystal, our unit cell was the primitive rhombohedral cell containing one Xe atom, as shown in Fig. 1, left panel. We sampled the Brillouin zone using a uniform 10x10x10 Monkhorst-Pack Γ-centered grid; a 14x14x14 grid gave a difference of less than 3 $\mu$eV in the total energy. Densities of states were calculated on a 12x12x12 grid. We used an energy cutoff $E_{\text{cutoff}} = 120$ Ry and set the cutoff energy for the augmentation charge density to eight (four) times this value (960 Ry (480 Ry)) for the US (NC) pseudopotentials. To compute the value of the lattice parameter for each of the functionals and van der Waals corrections, we computed the total energy for a series of fixed lattice parameters, then performed a parabolic fitting of the total energy as a function of the lattice parameter and took the minimum.

Following the liquid xenon MC calculations, we extracted representative supercells for the subsequent DFT calculations as follows: the number of Xe atoms in a cubic supercell of edge 24.88 a.u. was set to 30 to match the density of the liquid and cells containing Xe atoms closer than 6.6 a.u. on application of periodic boundary conditions were discarded.

For calculations of the electronic properties of liquid xenon, including the electron density in momentum space and its gradients, we used an energy cutoff value of $E_{\text{cutoff}} = 120$ Ry. Because of the absence of a crystal structure, we performed Γ-point calculations as for the atomic case.

# B  Physics of the various van der Waals corrections

Two main classes of corrections exist to include van der Waals interactions in DFT.

A first type of correction adds a fully non-local part to the XC functional. The vdW-DF XC functional belongs to this family [69, 70]. Starting from the GGA functional revPBE [71], it modifies the correlation energy by incorporating an additional geometry-dependent density-density interaction [70], and has been shown to reduce the GGA overestimation of lattice parameters [69, 70].

vdW-DF2 [72] further improves the accuracy of this method by building on the PW86 XC functional [73], which describes the region near equilibrium separations more accurately than the too repulsive revPBE. In addition, the non-local kernel of the functional is obtained via a large-N asymptote expansion of the exchange, which describes intra-molecular charge correlations better than the expansion for the slowly varying electron gas used in revPBE [72]. When tested on several molecular duplexes from the quantum chemistry S22 set [74, 75], it yielded binding energies, equilibrium distances and potential energy curves in good agreement with precise quantum-chemistry calculations [72], solving typical problems of vdw-DF such as the underestimation of hydrogen bond strengths and the overestimation of equilibrium distances [72].

Also in this class, the VV10 non-local functional [76], and its revised form for plane waves rVV10 [77], use a simple analytical form for the non-local correlation energy term with only the electron density as input. Thanks to a convenient form of the non-local correlation kernel, written in terms of the plasma frequency and the local band gap, important and well-justified spatial asymptotes for $R \rightarrow 0$, $R \rightarrow \infty$ as well as the uniform density limit are satisfied [76]. The obtained functional can be computed with simpler mathematical operations than its predecessors and does not present any explicit dependence on Kohn-Sham orbitals [76]. rVV10 excellently describes the binding energy of the Ar dimer, as well as the lattice parameters and bulk moduli of various solids with different bonding nature, from ionic to metallic. When tested against the quantum chemistry S22 set, it gave errors in binding energies typically smaller than 1 kcal/mol [77]. Compared to vdW-DF2, rVV10 further reduces the overestimation of the lattice parameters [77].

An alternative approach to accounting for dispersion forces is to add a correction to the DFT energy without explicitly modifying the underlying XC functional. This makes these corrections, generally, less complex than the non-local functionals previously mentioned. One example of this second class is the DFT-D functional [78], which adds two-body interatomic potentials of the form $C_6 R^{-6}$, with the coefficients obtained empirically. It has been widely applied to a variety of materials and properties, and has shown good results, outperforming GGA, as well as meta-GGA and hybrid functionals [78, 79], in particular yielding an error in the equilibrium bond length of the xenon dimer of the order of 0.1% [80]. Its revised version DFT-D3 [81] includes three-body dispersion terms, with the cutoff distances and coefficients of the correction calculated ab-initio. This revision improves the accuracy of DFT-D in describing extended van der Waals-dominated systems, such as graphite, where DFT-D is generally overbinding [81].

Finally, the Tkatchenko-Scheffler (TS) method [82] computes the molecular $C_6$ van der Waals coefficients from the atomic polarizabilities and ground-state charge density determined via DFT and time-dependent DFT. The many-body dispersion method [83] (mbd-vdw) then builds on the TS method by including $n$-body terms with $n > 2$ and incorporating electrostatic screening at large distances, which affects the polarizabilities of the individual atoms in molecules and solids. It gives good accuracy for the binding energies of van der Waals-dominated molecules in the usual S22 database, as well as for cohesive energies of molecular crystals [83].

## C    Calculation of the refractive index

We extracted the refractive index from the dielectric tensor computed using density functional perturbation theory within `Quantum Espresso`. We varied the frequency of the electric field perturbation (corresponding to the interband transition energy) between 0 and 80 eV and computed the dielectric function at 1000 evenly spaced frequencies over this range. Note that this treatment considers only vertical ($q = 0$) transitions.

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
