# Peer review of "Electronic structure of liquid xenon in the context of light dark matter direct detection"

_SciPost Physics, doi:SciPost Phys. 19, 064 (2025)_

## Round 1 · Referee Report · Anonymous (Referee 1) · 2025-5-27

Strengths

1- The authors clearly stated the problem which they wanted to address. Dark matter (DM) detection experiments are designed by considering the range of the masses of assumed DM particles. For heavy candidates, above GeV, recoil of the atomic nucleus is relevant. However, for light mass DM candidate particles, ie., lighter than 1 GeV, DM - electron interactions are important. In regard to the existing detectors containing liquid xenon, a formalism is necessary to compute scattering rates for the electrons of the liquid xenon. The introduction is very good at explaining both the problem and the approach taken by the authors to investigate it.

2- The authors obtained an expression for the dark matter (DM) - electron interaction rate which can be used either for an isolated atom or for the liquid state. This provides a convenient way to compare the effect of different targets on the interaction rate.

3- In the case of an isolated atom, Hartree-Fock type approaches can be sufficient. However, for the description of the electronic structure in condensed phases, the preferred state-of-the-art methods are based on density functional theory (DFT). Hence, a comparison of the atom versus liquid state requires also a comparison of different electronic structure methodologies. Furthermore, the DFT implementations are not unique, there are different exchange-correlation functional parametrizations, different ways of incorporating van der Waals interactions, and the issues of DFT+U methods and the treatment of spin-orbit coupling .. all these imply a wide spectrum of computational tools available for such a study. The authors provide a comparison of many such methods and justify their choice. They also provide a comparison of standard Roothan-Hartree-Fock method and the chosen DFT method.

Weaknesses

1- I have not found a major weakness. There are two issues; however, the authors mention and discuss them explicitly and identify them in the list of future improvements in the outlook paragraphs. These issues are the employment of a pseudopotential method and a better representation of the final electron wave function.

Report

The manuscript reports the results of a theoretical study on if the electronic structure of liquid xenon matters significantly in the description of the dark matter - target material interactions. Overall, many aspects of the problem were investigated carefully. The introduced approximations were explained, and made plausible. Such detailed discussions increase the reliability of the obtained results. The text is clear, easy to follow. The manuscript satisfies all the six general acceptance criteria.

Requested changes

1- The implementation of the DFT+U method requires the removal of the effects already included through the exchange-correlation functional in the DFT part. This, so-called double count correction, can be done in various ways, fully-localized limit, around the mean field, etc. This information is not provided here. Please provide it.

2- Page 8, first column: "LDA PZ (lower panel)" --> LDA PZ (left panel)

3- Page 8, second column: In the last paragraph of the subsection III.C.1, I think it will be better if the theoretical lattice parameter obtained by the preferred functional is explicitly stated. In the current text, this information (11.42 au) is given in Fig. 5 (figure and caption). The reason is that in this section the lattice parameter plays the decisive role, but at the end the reader does not know the best value. This information comes after two pages.

4- Page 10, first column: "this density density matches": remove one of the density words.

Recommendation

Publish (surpasses expectations and criteria for this Journal; among top 10%)

  • validity: top
  • significance: top
  • originality: high
  • clarity: top
  • formatting: excellent
  • grammar: perfect

Author:  Nicola Spaldin  on 2025-07-31  [id 5696]

(in reply to Report 1 on 2025-05-27)

Please see attached file

Attachment:

Response_to_Referees.pdf

---

## Round 1 · Referee Report · Anonymous (Referee 2) · 2025-5-28

Strengths

1—Clearly written.
2—Well organized structure.
3—Clear motivation of the work
4—Complete theoretical overview of dark matter induced ionization in xenon detectors.
5—Good comparisons to experiment to justify DFT approximations
6—interesting dark matter induced xenon ionization rate results

Weaknesses

1—The writing is overly verbose in places
2—Detailed electronic structure analysis is not consistently strong throughout
3—The discussion of the dark matter induced ionization rates in Sec IV C is underdeveloped compared to the rest of the manuscript.

Report

The manuscript titled “Electronic structure of liquid xenon in the context of light dark matter direct detection” by R. Catena et al. perform an in-depth study of the electronic structure and electronic density of both liquid and solid xenon, and estimate the dark matter scatting rate with an eye towards the XENON10 and XENON1T experiments. Various properties such at the 4d levels were found to be sensitive to the treatment of exchange-correlation effects. Moreover, a marked change in the 5p band width between solid/liquid phases influences the dark matter scatting rate. Overall, I find this work quite comprehensive, readable --though verbose at times--, and provide a good baseline for future studies in this field. Before I can accept this article, I have a few questions and comments/suggestions for the authors below.

(A) When the authors say (first paragraph of Sec. III) “…use of a plane-wave basis set for the expansion of the Kohn-Sham states is particularly convenient for ensuring convergence.” What is meant? Convergence as in accuracy? Number of SCF steps needed? Each basis/implementation has their own pros/cons, so it would be good to be clear about the methods used and why so that non-experts in electronic structure can clearly get the point.

(B) Since the Hubbard U is an empirical fix, I am curious as to the performance of metaGGA (e.g. SCAN, HSE, mBJ, …) functionals and/or MBPT corrections (e.g. GW) on the xenon 4d states? Furthermore, references to previous work in this section are quite slim. Is there truly very little work in this area? Are there no other theoretical calculations to compare with?

(C) For the lattice constants, yes LDA over binds and PBE under binds so it makes sense the vdW correction helps to fix PBE, but is it for the right reason? Would the vdW correction be minimal for a functional that captures a good lattice parameter out-of-the-box? Or is this a general limitation of semi-local functionals, making their description of the subtle long-range interactions between xenon atoms out of reach? Since I see this work setting the baseline in this area, I think it would be good to fully tease out the key problem areas for electronic structure modeling to tackle in this field.

(D) Can the authors give some reference(s) to justify the appropriateness of the Lennard-Jones potential for xenon? There are a number of pairwise, many-body potentials, and specialized models in the field, so some reasoning as to why Lennard-Jones would be good to call out. Moreover, are the parameters used similar to those that would be obtained with DFT?

(E) In Fig. 5, can the authors offer some insights into why the peak in liquid phase theoretical RDF at 15 [a.u.] appears to be shifted to higher larger distances? Also, is there a physical reason why the peak in the experimental RDF at $\sim$13 [a.u.] is missing in the liquid phase theoretical results?

(F) It will be good to present the calculated dielectric function to facilitate more direct comparison with the electronic structure results.

(G) Since a number of different functional+vdw+Hubbard corrections were discussed in Sec. III, it is unclear what was actually used for the data in Sec. IV. Please call this out in text and clearly specify this in the relevant figure captions.

(H) Since Sec. III E and F feel auxiliary to the flow of the text, I suggest moving Sec. III E and F to an appendix.

(I) When comparing/analyzing the electronic densities, have the authors compared against an all-electron code, such as Wien2k? In such codes there is no pseudopotential problem and they could provide a good benchmark to ensure the quality of the planewave calculations. This will be important to certify the important high momentum regimes of the density. Also, currently Fig. 9 does not add much to the discussion, but adding comparisons to other pseudopotentials and/or all-electron codes would enrich the results.

(J) When modeling the final states (page 15 bottom left-hand column), this discussion appears to be similar to those that occurred in the 1970s surrounding the modeling of ARPES spectra, i.e. pros/cons of the 1 step and 3 step model. Could this field learn from these previous discussions? Additionally, in this connection, calculated ARPES spectra is highly sensitive to the surface potential. Can the authors speak to this issue in the context of xenon detectors a bit more and how it might effect the utilization of the theoretical predictions such as those in Fig. 14 for XENON10 and XENON1T experiments?

(K) Why is there a kink in the blue lines of Fig. 14 (a)?

(L) The text in a number of places throughput the manuscript is heavy, I suggest editing to help with the flow.

Requested changes

In addition to addressing questions (A)-(L) in the Report section, please fix the minor issues below:

1) In the abstract, can the authors be specific as to the numerical range of ‘low’ when they state “…5p levels induced by the liquid phase is relevant only at low dark matter masses…”? Similarly, when they say “For most of the probable mass range the energies…”, can ‘probable’ be defined. Adding the mass ranges explicitly will help the reader at a glance determine the relevance of this work to their own interstates without digging in the text.

2) Footnote 3, change ‘consistently’ to ‘consistent’

3) In figure 5, it will be good to specify the temperature at which the experimental data was collected.

4) First paragraph in the left-hand column of Page 10, it says “We obtain an fcc structure and the sharp Dirac-delta-like peaks in the RDF shown with purple solid lines in Fig. 5.” I believe the authors mean as shown in blue solid lines.

5) Since there are many types of ‘material response functions’ in condensed matter, is there a better name for W? For instance, when modeling dark matter scatting in solids authors have used similar terms when discussing the dielectric response, which is different from the meaning here.

6) I find it strange to indicate vdw-DF, vdw-DF2, and rVV10 on the horizontal axis of Fig. 3 (a) but not plot anything. Since QE does not have this combination of treatments implemented, I suggest just stating it in the text (as was done) and leave them out of the figure.

Recommendation

Ask for major revision

---

## Round 2 · Referee Report · Anonymous (Referee 2) · 2025-8-3

Report

I thank the authors for their hard work and the revised manuscript. The current version satisfies my questions/comments and agree with leaving the few specific points to editorial discretion. I recommend the article for publication.

Recommendation

Publish (easily meets expectations and criteria for this Journal; among top 50%)

---

## Round 2 · Author Response

Response to the Report of Referee 1 – 2502.02965v1

The manuscript reports the results of a theoretical study on if the electronic structure of liquid xenon matters significantly in the description of the dark matter - target material interactions. Overall, many aspects of the problem were investigated carefully. The introduced approximations were explained, and made plausible. Such detailed discussions increase the reliability of the obtained results. The text is clear, easy to follow. The manuscript satisfies all the six general acceptance criteria. Publish (surpasses expectations and criteria for this Journal; among top 10%) We thank the referee for the kind and positive evaluation of our work.

Requested changes Referee 1 – 2502.02965v1

1- The implementation of the DFT+U method requires the removal of the effects already included through the exchange-correlation functional in the DFT part. This, so-called double count correction, can be done in various ways, fully-localized limit, around the mean field, etc. This information is not provided here. Please provide it.

We thank the referee for pointing out that we omitted to provide this information. We use the simplified rotationally invariant approach of Dudarev et al. with an effective Hubbard parameter, Uef f= U−J. We have added this statement to the text and included the appropriate reference. We have also changed U to Uef f where appropriate throughout the manuscript.

2- Page 8, first column: ”LDA PZ (lower panel)” −→LDA PZ (left panel) Done

3- Page 8, second column: In the last paragraph of the subsection III.C.1, I think it will be better if the theoretical lattice parameter obtained by the preferred functional is explicitly stated. In the current text, this information (11.42 au) is given in Fig. 5 (figure and caption). The reason is that in this section the lattice parameter plays the decisive role, but at the end the reader does not know the best value. This information comes after two pages.

We agree with the referee and we have expanded the text in page 8 as ...we select the DFT-D correction to the PBE XC functional (with calculated lattice parameter of 11.42 a.u.) as our choice to treat van der Waals interactions and use it in the subsequent calculations.

4- Page 10, first column: ”this density density matches”: remove one of the density words. Done

We again thank the referee for the careful evaluation of the findings presented herein and for recommending our manuscript for publication.

Response to the Report of Referee 2 – 2502.02965v1

The manuscript titled “Electronic structure of liquid xenon in the context of light dark matter direct detection” by R. Catena et al. perform an in-depth study of the electronic structure and electronic density of both liquid and solid xenon, and estimate the dark matter scatting rate with an eye towards the XENON10 and XENON1T experiments. Various properties such at the 4d levels were found to be sensitive to the treatment of exchange-correlation effects. Moreover, a marked change in the 5p band width between solid/liquid phases influences the dark matter scatting rate. Overall, I find this work quite comprehensive, readable –though verbose at times–, and provide a good baseline for future studies in this field. Before I can accept this article, I have a few questions and comments/suggestions for the authors below.

We thank the referee for the positive evaluation of our work and hope that after addressing all of the raised points below, the article will be ready for publication.

(A) When the authors say (first paragraph of Sec. III) “. . . use of a plane-wave basis set for the expansion of the Kohn-Sham states is particularly convenient for ensuring convergence.” What is meant? Convergence as in accuracy? Number of SCF steps needed? Each basis/implementation has their own pros/cons, so it would be good to be clear about the methods used and why so that non-experts in electronic structure can clearly get the point.

Indeed, we tried to write this paper so that it is accessible to both the electronic structure and astroparticle physics communities, and we appreciate the referee pointing out where we have been unclear. The plane-wave basis is particularly convenient because the total energy is variational in the number of plane waves, so convergence can be tested simply by increasing the number of plane waves included in the calculation. Convergence is achieved when the change in the relevant property (usually in total energy) on including more plane waves is less than the desired accuracy. We made the following change in the manuscript: "...use of a plane-wave basis set for the expansion of the Kohn-Sham states is particularly convenient, since convergence of the total energy can be straightforwardly assessed by increasing the number of plane waves."

(B) Since the Hubbard U is an empirical fix, I am curious as to the performance of metaGGA (e.g. SCAN, HSE, mBJ, . . . ) functionals and/or MBPT corrections (e.g. GW) on the xenon 4d states? Furthermore, references to previous work in this section are quite slim. Is there truly very little work in this area? Are there no other theoretical calculations to compare with?

As discussed in the manuscript, the unphysically high energy of the core states in DFT is a result of the spurious self interaction. Since metaGGA functionals in general do not introduce a self-interaction correction we do not expect them to offer a systematic improvement in the core energies. We have tested the behavior of HSE06 on semi-core states and established that, while tuning the amount of exact exchange allows one to obtain the binding energies reported in XPS, this amount of exact exchange does not give the experimental band gap, and does not have any particular physical justification. Regarding the GW method, our experience with other materials is that the GW tends to shift core states to even higher energy (i.e. in the wrong direction compared to experiment), particularly when the parameters in the calculation are tuned to optimize the band gap (which is the usual realm of application of GW). Since neither HSE nor GW give a systematic, physically justified improvement, and in addition would be rather computationally expensive for our liquid supercells, we choose not to use them. We were also surprised at the lack of earlier first-principles calculations of electronic energy levels and lattice constants (for the crystalline solid) for Xe, and would be happy if the referee knows of examples that we have missed in the literature. We note that there are a number of simulations of shock physics using ab initio molecular dynamics that we did not include in our discussion as we did not find them relevant for this study.

(C) For the lattice constants, yes LDA over binds and PBE under binds so it makes sense the vdW correction helps to fix PBE, but is it for the right reason? Would the vdW correction be minimal for a functional that captures a good lattice parameter out-of-the-box? Or is this a general limitation of semi-local functionals, making their description of the subtle long-range interactions between xenon atoms out of reach? Since I see this work setting the baseline in this area, I think it would be good to fully tease out the key problem areas for electronic structure modeling to tackle in this field.

We agree with the referee that the inclusion of van der Waals corrections will either reduce the lattice constants if non-local interactions are relevant in the material, or have no effect on the lattice constant if they are not. If a local or semi-local functional were to predict the correct lattice constant for Xe, we would in fact argue that it could not be capturing the correct physics and that the agreement would be fortuitous; subsequent incorporation of vdW corrections would then indeed overbind.

(D) Can the authors give some reference(s) to justify the appropriateness of the Lennard-Jones potential for xenon? There are a number of pairwise, many-body potentials, and specialized models in the field, so some reasoning as to why Lennard-Jones would be good to call out. Moreover, are the parameters used similar to those that would be obtained with DFT?

There are a number of studies arguing for the use of Lennard-Jones potential for noble liquids (not under abnormally high pressures) such as [1-4] which also compare to other, more involved forms of potential. Its simplicity and a large track record of successful uses has led us to choose this form of interatomic potential. We expect that our final results would be largely unchanged for an alternative potential. [1] Solana Quir´os, Jos´e Ram´on and Akhouri, Binay, Thermodynamic Properties of Ar, Kr and Xe from a Monte Carlo- Based Perturbation Theory with an Effective Two-Body Lennard-Jones Potential. Available at SSRN: https://ssrn.com/abstract=4170658 or http://dx.doi.org/10.2139/ssrn.4170658 [2] G. K. Horton and J. W. Leech, “On the statistical mechanics of the ideal inert gas solids,” Proc. Phys. Soc. 82, 816–854 (1963) [3] Asuka J. Iwasaki, Marcin Kirsz, Ciprian G. Pruteanu, and Graeme J. Ackland The Journal of Physical Chemistry Letters 2025 16 (6), 1559-1566 DOI: 10.1021/acs.jpclett.4c03272 [4] S. Stephan, J. Staubach, H. Hasse. (2020). Review and Comparison of Equations of State for the Lennard-Jones Fluid. Fluid Phase Equilibria. 523. 112772. 10.1016/j.fluid.2020.112772.

(E) In Fig. 5, can the authors offer some insights into why the peak in liquid phase theoretical RDF at 15 [a.u.] appears to be shifted to higher larger distances? Also, is there a physical reason why the peak in the experimental RDF at 13 [a.u.] is missing in the liquid phase theoretical results?

We thank the referee for raising this issue. We chose to include this set of experimental data, which is from The Structure of Liquid Xenon, J. A. Campbell and J. H. Hildebrand, J. Chem. Phys. 11, 334–337 (1943), as it is closest (although not equal) in pressure and temperature to the case we consider. More modern diffraction studies (see for example Phys. Rev. B 153, 229 (1967) and Phys. Rev. B 45, 4605 (1992)) do not show the peak at 12 a.u., nor do other molecular dynamics simulations, using either Lennard-Jones or other potentials. There is also considerable spread in the literature for the broad peak centered around 15 a.u. We have added a discussion of this point, including the references, to the manuscript.

(F) It will be good to present the calculated dielectric function to facilitate more direct comparison with the electronic structure results.

Actually, we chose to show the calculated refractive index rather than the dielectric constant because of the availability of experimental data for comparison. While we could include our calculated dielectric function as well, we find it redundant as it is directly related to the refractive index, but we are happy to leave this to an editorial decision.

(G) Since a number of different functional+vdw+Hubbard corrections were discussed in Sec. III, it is unclear what was actually used for the data in Sec. IV. Please call this out in text and clearly specify this in the relevant figure captions.

We agree with the referee and have modified the text.

(H) Since Sec. III E and F feel auxiliary to the flow of the text, I suggest moving Sec. III E and F to an appendix.

Actually, we would prefer to keep the pseudopotential discussion in the main text as we use it to illustrate the relative importance of different approximations used in the calculations. We have tried to clarify in the manuscript our motivation for this section. Section F could move to an Appendix, although it is so short we don’t find it to be an interruption. We leave this to an editorial decision.

(I) When comparing/analyzing the electronic densities, have the authors compared against an all-electron code, such as Wien2k? In such codes there is no pseudopotential problem and they could provide a good benchmark to ensure the quality of the planewave calculations. This will be important to certify the important high momentum regimes of the density. Also, currently Fig. 9 does not add much to the discussion, but adding comparisons to other pseudopotentials and/or all-electron codes would enrich the results.

We agree that Figure 9 is occupying too much real estate and have reduced its size in the new version of the manuscript. With this comment the referee addresses exactly a main point of our work: Since the pseudopotential approximation does not include the high-momentum components of the electronic wavefunctions it can not be used to calculate scattering rates for high momentum transfer events. We emphasize that we do not use our pseudopotential results in this regime. One possibility for the high-momentum-transfer regime is indeed to use instead an all-electron code as the referee suggests. (Note, however, that an all-electron code does not in general ensure a higher quality calculation!) The proposal that we introduce here, which is a much cheaper alternative, is to use the well-established all-electron wavefunctions for the Xe atom, weighted by the pseudopotential DFT-calculated density of states for the liquid, to construct the initial electronic structure of the liquid. We have expanded the discussion section of the manuscript to clarify this issue further.

(J) When modeling the final states (page 15 bottom left-hand column), this discussion appears to be similar to those that occurred in the 1970s surrounding the modeling of ARPES spectra, i.e. pros/cons of the 1 step and 3 step model. Could this field learn from these previous discussions? Additionally, in this connection, calculated ARPES spectra is highly sensitive to the surface po- tential. Can the authors speak to this issue in the context of xenon detectors a bit more and how it might effect the utilization of the theoretical predictions such as those in Fig. 14 for XENON10 and XENON1T experiments?

Indeed we have been following closely the recent developments in the modeling of the final state in ARPES and have added a mention of this as well as the most relevant reference to the Conclusion section. In analogy with the theoretical modeling of ARPES spectra, our description of the DM-induced ionization rate in xenon detectors can be thought of as a “phenomenological” three-step model. Specifically, Step 1 (i.e., the primary electron emission) is captured by Eq. (1). Steps 2 (i.e., electron transport in the liquid xenon phase) and 3 (i.e. transition from the liquid to the gaseous xenon phase and S2 signal production) are encoded in the binomial and gaussian probabilities, as well as in the detector efficiency in Eq. (36). Effects related to the liquid xenon surface potential, which are important in Step 3, are accounted for by multiplication by the detector efficiency, which effectively models the probability of extracting an electron from the liquid to the gaseous xenon phase.

(K) Why is there a kink in the blue lines of Fig. 14 (a)?

The ionization events recorded in XENON10 have been divided into intervals, each associated with a different range for S2, i.e., the observable number of photoelectrons in an ionization event. The 90% C.L. exclusion limits reported in Fig. 14 have been obtained by requiring that the joint probability of DM producing numbers of ionization events larger than those observed in each S2 bin is 90%. In the heavy mediator scenario, different S2 bins give the largest contribution to this joint probability for DM masses higher than or below about 50 MeV. This explains the change of slope in the blue curve in the left panel of Fig. 14. This effect is also discussed in Ref. [10] (see Fig. 7). We have commented on this in the caption of Fig. 14 as: "The change of slope of exclusion limits for heavy mediator in XENON10 is due to a division of the recorded events into two intervals depending on the recorded number of photoelectrons, which are kinematically accessible at different mass scales."

(L) The text in a number of places throughput the manuscript is heavy, I suggest editing to help with the flow.

We have worked through the manuscript again with the help of suggestions from the overleaf AI editor and hope that this has improved the linguistic accessibility.

Requested changes Referee 2 – 2502.02965v1

In addition to addressing questions (A)-(L) in the Report section, please fix the minor issues below: 1) In the abstract, can the authors be specific as to the numerical range of ‘low’ when they state “. . . 5p levels induced by the liquid phase is relevant only at low dark matter masses. . . ”? Similarly, when they say “For most of the probable mass range the energies. . . ”, can ‘probable’ be defined. Adding the mass ranges explicitly will help the reader at a glance determine the relevance of this work to their own interstates without digging in the text.

We agree with the referee and have modified the text of the abstract as "We find that the broadening of the 5p levels induced by the liquid phase is relevant only for dark matter masses below 6 MeV, where it increases the ionisation rate relative to that of the isolated atom." Furthermore, we have clarified the latter sentence as "For most of the explored mass range the energies. . ."

2) Footnote 3, change ‘consistently’ to ‘consistent’ Done

3) In figure 5, it will be good to specify the temperature at which the experimental data was collected.

We agree with the referee and have added this information to the caption of Fig. 5 as "compared to the experimentally measured distribution at 183 K (yellow dots)"

4) First paragraph in the left-hand column of Page 10, it says “We obtain an fcc structure and the sharp Dirac-delta-like peaks in the RDF shown with purple solid lines in Fig. 5.” I believe the authors mean as shown in blue solid lines.

The referee is correct, we have fixed this typo.

5) Since there are many types of ‘material response functions’ in condensed matter, is there a better name for W? For instance, when modeling dark matter scatting in solids authors have used similar terms when discussing the dielectric response, which is different from the meaning here.

We have chosen the nomenclature in this manuscript to reflect the usual terms used by the dark matter community for the past decade. We agree that this is not the standard notation in the condensed matter community, but to be consistent with our previous works and with other works in the field, we prefer to keep this naming convention. We hope that the term is sufficiently well-defined in the text to avoid confusion.

6) I find it strange to indicate vdw-DF, vdw-DF2, and rVV10 on the horizontal axis of Fig. 3 (a) but not plot anything. Since QE does not have this combination of treatments implemented, I suggest just stating it in the text (as was done) and leave them out of the figure.

We find the comparison between the left and right panels more straightforward if the x axes are kept consistent, but are happy to leave this to an editorial decision.

We again thank the referee for the careful evaluation of the findings presented herein. We hope that after addressing all the points raised above and modifying the manuscript accordingly, this version is ready for publication in SciPost.

---

## Round 2 · List of Changes

Please see list of changes in the response to the referees above.

---

## Editorial Decision

published